# Video PreTraining (VPT): Learning to Act by Watching Unlabeled Online Videos

**Bowen Baker**[*†]
bowen@openai.com

**Ilge Akkaya**[*†]
ilge@openai.com

**Peter Zhokhov**[*†]
peterz@openai.com

**Joost Huizinga**[*†]
joost@openai.com

**Jie Tang**[*†]
jietang@openai.com

**Adrien Ecoffet**[*†]
adrien@openai.com

**Brandon Houghton**[*†]
brandon@openai.com

**Raul Sampedro**[*†]
raulsamg@gmail.com

**Jeff Clune**[*†‡]
jclune@gmail.com

## Abstract

Pretraining on noisy, internet-scale datasets has been heavily studied as a technique for training models with broad, general capabilities for text, images, and other modalities.[1–6] However, for many sequential decision domains such as robotics, video games, and computer use, publicly available data does not contain the labels required to train behavioral priors in the same way. We extend the internet-scale pretraining paradigm to sequential decision domains through semi-supervised imitation learning wherein agents learn to act by watching online unlabeled videos. Specifically, we show that with a small amount of labeled data we can train an inverse dynamics model accurate enough to label a huge unlabeled source of online data – here, online videos of people playing Minecraft – from which we can then train a general behavioral prior. Despite using the native human interface (mouse and keyboard at 20Hz), we show that this behavioral prior has nontrivial zero-shot capabilities and that it can be fine-tuned, with both imitation learning and reinforcement learning, to hard-exploration tasks that are impossible to learn from scratch via reinforcement learning. For many tasks our models exhibit human-level performance, and we are the first to report computer agents that can craft diamond tools, which can take proficient humans upwards of 20 minutes (24,000 environment actions) of gameplay to accomplish.

## 1 Introduction

Work in recent years has demonstrated the efficacy of pretraining large and general foundation models[7] on noisy internet-scale datasets for use in downstream tasks in natural language[1–4], computer vision,[5,6,8] and multi-task models.[9] For sequential decision domains (e.g. robotics, game playing, and computer usage) where agents must repeatedly act within an environment, a wealth of data also exists on the web; however, most of this data is in the form of *unlabeled* video (i.e. without the actions taken at each frame), making it much less straightforward to train a behavioral prior in these

---

[*]This was a large effort by a dedicated team. Each author made huge contributions on many fronts over long time periods. All members were full time on the project for over six months. BB, IA, PZ, and JC were on the original VPT project team and were thus involved for even longer (over a year). Aside from those original team members, author order is random. It was also randomized between IA and PZ.

[†]OpenAI

[‡]University of British Columbia

36th Conference on Neural Information Processing Systems (NeurIPS 2022).

domains than it is in e.g. natural language. In a few rare settings, such as Chess, Go, and StarCraft, there already exist large datasets with action labels from various online platforms that researchers have used for imitation learning.[10,11] When large labeled datasets do not exist, the canonical strategy for training capable agents is reinforcement learning (RL),[12] which can be sample inefficient and expensive for hard-exploration problems.[13–19] Many virtual tasks, e.g. navigating websites, using Photoshop, booking flights, etc., can be very hard to learn with RL and do not have large, commonly available sources of labeled data.[20,21] In this paper, we seek to extend the paradigm of training large, general-purpose foundation models to sequential decision domains by utilizing freely available internet-scale unlabeled video datasets with a simple semi-supervised imitation learning method. We call this method Video PreTraining (VPT) and demonstrate its efficacy in the domain of Minecraft.

Existing semi-supervised imitation learning methods aim to learn with few or no explicit action labels; however, they generally rely on the policy's ability to explore the environment throughout training, making them susceptible to exploration bottlenecks.[22–26] Furthermore, most prior semi-supervised imitation learning work was tested in the relatively low data regime; because we experiment with *far* more data (∼70k hours of unlabeled video), we hypothesize that we can achieve good performance with a much simpler method, a trend that has proven true for pretraining in other modalities such as text.[1] In particular, given a large but unlabeled dataset, we propose generating pseudo-labels by gathering a small amount of labeled data to train an inverse dynamics model (IDM) that predicts the action taken at each timestep in a video. Behavioral cloning (BC) can require a large amount of data because the model must learn to infer intent and the distribution over future behaviors from only past observations. In contrast, the inverse dynamics modeling task is simpler because it is *non-causal*, meaning it can look at both past and future frames to infer actions. In most settings, environment mechanics are far simpler than the breadth of human behavior that can take place within the environment, suggesting that non-causal IDMs could require far less data to train than causal BC models. Using pseudo-labels generated from the IDM, we then train a model to mimic the distribution of behavior in the previously unlabeled dataset with standard behavioral cloning at scale, which does not require any model rollouts and thus does not suffer from any potential exploration bottlenecks in the environment. Finally, we show we can fine-tune this model to downstream tasks with either behavioral cloning or reinforcement learning.

We chose to test our method in Minecraft because it (a) is one of the most actively played games in the world[27] and thus has a wealth of online video data, (b) is an open-ended sandbox game with an extremely wide variety of potential things to do, build, and collect, making our results more applicable to real-world applications such as computer usage, which also tends to be varied and open-ended, and (c) has already garnered interest by the RL community as a research domain due to its complexity and correspondingly difficult exploration challenges.[28–32] In this work we use the native human interface for Minecraft so that we can (1) most accurately model the human behavior distribution and reduce domain shift between video data and the environment, (2) make data collection easier by allowing our human contractors to play the game without modification, and (3) eliminate the need to hand-engineer a custom interface

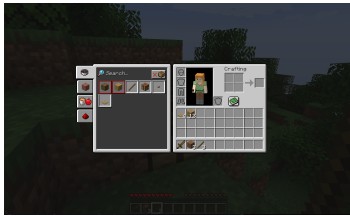

Figure 1: Example Minecraft crafting GUI. Agents use the mouse and keyboard to navigate menus and drag and drop items.

for models to interact with the environment. This choice means that our models play at 20 frames per second and must use a mouse and keyboard interface to interact with human GUIs for crafting, smelting, trading, etc., including dragging items to specific slots or navigating the recipe book with the mouse cursor (Fig. 1). Compared to prior work in Minecraft that uses a lower frame rate and constructs crafting and attacking macros,[31,33–35] using the native human interface drastically increases the environment's exploration difficulty, making most simple tasks near impossible with RL from scratch. Even the simple task of gathering a single wooden log while already facing a tree takes 60 consecutive attack actions with the human interface, meaning the chance for a naive random policy to succeed is $1/2^{60}$. While this paper shows results in Minecraft only, the VPT method is general and could be applied to any domain.

In Section 4 we show that the VPT foundation model has nontrivial zero-shot performance, accomplishing tasks impossible to learn with RL alone, such as crafting planks and crafting tables (tasks requiring a human proficient in Minecraft a median of 50 seconds or ∼970 consecutive actions). Through fine-tuning with behavioral cloning to smaller datasets that target more specific behavior

distributions, our agent is able to push even further into the technology tree, crafting stone tools (taking a human a median of 2.3 minutes or ∼2790 actions). Finally, fine-tuning via RL produces the most dramatic improvements: our agent is able to craft diamond tools, an unprecedented result in Minecraft made even more challenging by using the native human interface. This task requires a proficient human a median upwards of 20 minutes or ∼24000 actions. The main contributions of this work are (1) we are the first to show promising results applying semi-supervised imitation learning to extremely large, noisy, and freely available video datasets for sequential decision domains, (2) we show that such pretraining plus fine-tuning enables agents to solve tasks that were otherwise impossible to learn, (3) we show that labeled contractor data is far more efficiently used within the VPT method than it would be by directly training a foundation model from it and (4) we open source our contractor data, trained model weights, and Minecraft environment for future research into learning to act via semi-supervised imitation learning at scale.

## 2   Preliminaries and Related Work

Imitation learning methods[36–39] seek to construct a policy that accurately models the distribution of behavior in some dataset $D = \{(o_i, a_i)\}$, $i \in \{1...N\}$ of action-observation pairs. In order to roll out these policies in an environment, they must be *causal*, meaning they condition on observations from the current timestep $t$ and past timesteps only, i.e. $\pi \sim p(a_t|o_1...o_t)$. Imitation learning is simplest when demonstrations are labeled with corresponding actions. Imitating labeled trajectories has seen success in aerial vehicles,[40,41] self-driving cars,[42,43] board games,[10,44] and video games.[11,45]

When labeled demonstrations are not available, standard behavioral cloning will not work; however, there is a large body of work in imitating behavior from unlabeled demonstrations.[23] For instance, GAIL[24] constructs an adversarial objective incentivizing the trained policy to exhibit behaviors indistinguishable from those in the target dataset. Edwards et al.[46] propose to first learn a latent policy using unlabeled demonstrations and then map the learned latent actions to real actions using environment interaction. Peng et al.[47] use motion-capture methods to track agent positions in videos and then train RL agents to match these waypoints. Similarly, Behbahani et al.[48] and Aytar et al.[49] task a RL agent to match waypoints; however, their waypoints are embeddings from unsupervised feature learning models. Pathak et al.[50] and Nair et al.[51] train goal conditioned policies to take actions that move towards expert-provided goal states expressed as high dimensional visual waypoints. Most similar to our own work, Torabi et al.[25] simultaneously train (1) an inverse dynamics model (IDM),[52] which aims to uncover the underlying action between timesteps given observations of past and future timesteps, e.g. $p_{IDM}(a_t|o_t, o_{t+1})$, and (2) a behavioral cloning (BC) model on trajectories of observations labeled with the IDM. Data to train the IDM is collected by rolling out the BC model in the target environment such that both models improve in tandem. However, at any point in training if there are sequences in the dataset that the IDM performs poorly on, it requires that the BC model perform those or similar sequences in order for the IDM to improve and correctly label them. Therefore, if the BC model does not explore efficiently, it could severely slow down learning. In order to avoid this potential issue we opted for a simpler two-stage approach: we first train an IDM on a small number of labeled trajectories collected from human contractors (they play the game as would normally as we record their keypresses and mouse movements). Because human contractors reach most relevant parts of the state space, we can hold the IDM fixed throughout BC training.

Compared to most previous work in semi-supervised imitation learning, we experiment in the much more complex and open-ended environment of Minecraft. Minecraft is a voxel-based 3D video game that, due its popularity and wide variety of mechanics, has attracted a vast amount of RL research.[28,29,31–35,53–61] A large body of work focuses on small, custom-made Minecraft worlds with tasks such as navigation,[54,61] block placing,[55,56] instruction following,[59,60] combat,[57] and others.[29,32,58] Work operating in the massive, randomly generated environments of Minecraft itself has included hill climbing,[53] automated curriculum learning[31] and, most closely related to the RL experiments presented in Sec. 4.4, diamond mining.[28,33–35] However, to the best of our knowledge, there is no published work that operates in the full, unmodified human action space, which includes drag-and-drop inventory management and item crafting.

## 3   Methods

**Inverse Dynamics Models (IDM)**    VPT, illustrated in Figure 2, requires we first collect a small amount of labeled contractor data with which to train an inverse dynamics model $p_{IDM}(a_t|o_{1...T})$,

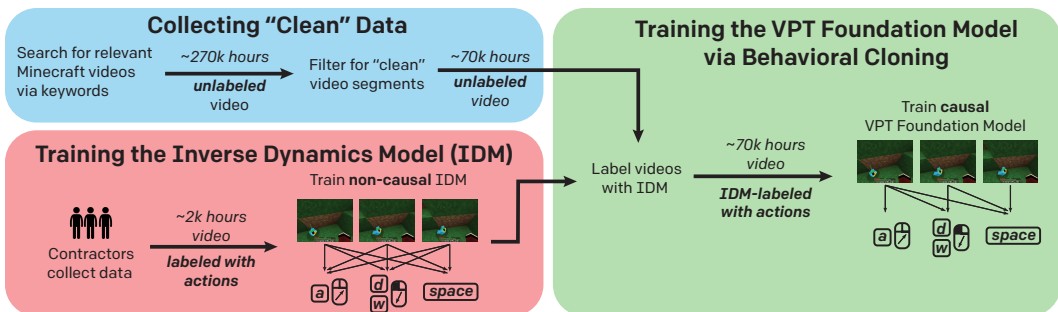

Figure 2: Video Pretraining (VPT) Method Overview.

which seeks to minimize the negative log-likelihood of an action at timestep $t$ given a trajectory of $T$ observations $o_t : t \in [1...T]$. In contrast to an imitation learning policy, the IDM can be non-causal, meaning its prediction for $a_t$ can be a function of both past and *future events*, i.e. $o_{t'>t}$. Compared to the behavioral cloning objective of modeling the distribution of human intent given past frames only, we hypothesize that inverting environment dynamics is easier and more data efficient to learn. Indeed, Sec. 4.1 will show that the IDM objective is much easier to learn, and furthermore Sec. 4.6 will show that with very little labeled data (as few as 100 hours) we can train a fairly accurate IDM. This IDM can be used to label online videos, providing the large amount of data required for the harder task of behavioral cloning. See appendices D and B for IDM training and data collection details.

**Data Filtering** We gather a large dataset of Minecraft videos by searching the web for related keywords (Appendix A). Online videos often (1) include overlaid artifacts, such as a video feed of the player's face, channel logos, watermarks, etc., (2) are collected from platforms other than a computer with different gameplay, or (3) are from different game modes, e.g. in Minecraft we only want "survival mode" where players start from scratch and must gather or craft all their items. We call data "clean" if it does not contain visual artifacts and is from survival mode, and call all other data "unclean." With enough data, a large enough model, and enough training compute, a BC model trained on both unclean and clean videos would likely still perform well in a clean Minecraft environment. However, for simplicity and training compute efficiency, we choose to filter out unclean segments of video (note that a video may contain both clean and unclean segments). We do this by training a model to filter out unclean segments using a small dataset (8800) of images sampled from online videos labeled by contractors as clean or unclean. We did not tune this process as it is fairly standard; see Appendix A.2 for more details and ablations showing data cleaning is beneficial.

**VPT Foundation Model** We train a foundation model with standard behavioral cloning, i.e. minimizing the negative log-likelihood of actions predicted by the IDM on clean data. For a particular trajectory of length $T$ we minimize

$$\min_{\theta} \sum_{t \in [1...T]} -\log \pi_\theta(a_t | o_1, \ldots, o_t), \text{ where } a_t \sim p_{\text{IDM}}(a_t | o_1, \ldots, o_t, \ldots, o_T) \quad (1)$$

As we will see in the following sections, this model exhibits nontrivial zero-shot behavior and can be fine-tuned with both imitation learning and RL to perform even more complex skills.

# 4 Results

## 4.1 Performance of the Inverse Dynamics Model

The IDM architecture is comprised primarily of a temporal convolution layer, a ResNet[63] image processing stack, and residual unmasked attention layers, from which the IDM simultaneously predicts keypresses and mouse movements (see Appendix D for IDM architecture and training details). A key hypothesis behind our work is that IDMs can be trained with a relatively small amount of labeled data. While more data improves both mouse movement and keypress predictions, our best IDM trains on only 1962 hours of data (compared to the $\sim$70k hours of clean data we collected from the internet) and achieves 90.6% keypress accuracy and a 0.97 $R^2$ for mouse movements evaluated on a held-out validation set of contractor-labeled data (Figure 3 left).

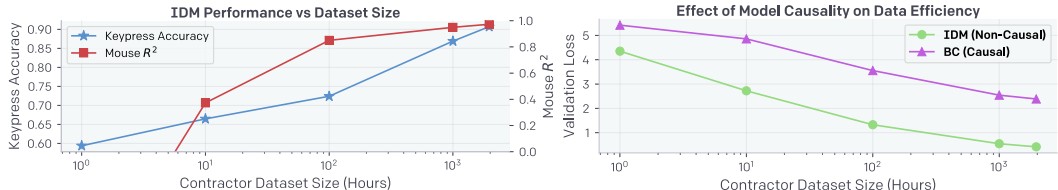

Figure 3: **(Left)** IDM keypress accuracy and mouse movement $R^2$ (explained variance[62]) as a function of dataset size. **(Right)** IDM vs. behavioral cloning data efficiency.

Figure 3 (right) validates our hypothesis that IDMs are far more data efficient than BC models, likely because inverting environment mechanics is far easier than modeling the entire distribution of human behavior. The IDM is two orders of magnitude more data efficient than a BC model trained on the same data and improves more quickly with more data. This evidence supports our hypothesis that it is more effective to use contractor data within the VPT pipeline by training an IDM than it is to train a foundation model from contractor data directly (Sections 4.5 and 4.6 provide additional evidence). Due to their data efficiency, training an IDM uses a negligible fraction of the overall compute needed to train a VPT model.

## 4.2 VPT Foundation Model Training and Zero-Shot Performance

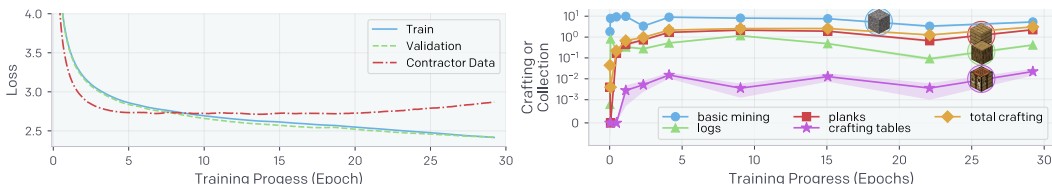

Figure 4: **(Left)** Training and validation loss on the `web_clean` internet dataset with IDM pseudo-labels, and loss on the main IDM contractor dataset, which has ground-truth labels but is out-of-distribution (see text). **(Right)** Amount a given item was collected per episode averaged over 2500 60-minute survival episodes as a function of training epoch, shaded with the standard error of the mean. Basic mining refers to collection of dirt, gravel, or sand (all materials that can be gathered without tools). Logs are obtained by repeatedly hitting trees for three seconds, a difficult feat for an RL agent to achieve as we show in Sec. 4.4. Planks can be crafted from logs, and crafting tables crafted from planks. Crafting requires using in-game crafting GUIs, and proficient humans take a median of 50 seconds (970 consecutive actions) to make a crafting table.

We now explore the emergent behavior learned by a behavioral cloning policy trained on an extremely large, but noisy, internet dataset labeled with our IDM. To collect the unlabeled internet dataset, we searched for publicly available videos of Minecraft play with search terms such as "minecraft survival for beginners." These searches resulted in ∼270k hours of video, which we filtered down to "clean" video segments yielding an *unlabeled* dataset of ∼70k hours, which we refer to as `web_clean` (Appendix A has further details on data scraping and filtering). We then generated pseudo-labels for `web_clean` with our best IDM (Section 3) and then trained the VPT foundation model with behavioral cloning. Preliminary model scaling experiments suggested that our model could benefit from 30 epochs of training and that a 0.5 billion parameter model was required to stay in the efficient learning regime[64] for that training duration (Appendix H shows results comparing model size and the benefit of scaling to 0.5B parameters), which took ∼9 days on 720 V100 GPUs.

We evaluate our models by measuring validation loss (Fig. 4, left) and rolling them out in the Minecraft environment. Unless otherwise noted, in all environment evaluations we spawn agents in a standard survival mode game where they play for 60 minutes, i.e. 72000 consecutive actions, and we plot the mean and shade the standard error of the mean for various game statistics such as crafting and collection rates (Fig. 4, right). The VPT foundation model quickly learns to chop down trees to collect logs, a task we found near impossible for an RL agent to achieve with the native human interface (Sec. 4.4). It also learns to craft those logs into wooden planks and then use those planks

to craft a crafting table, which are required to unlock most other technology in the game and take a human proficient in Minecraft approximately 50 seconds (970 consecutive actions) to collect. While these behaviors are fairly complex in the native human action space, the VPT foundation model crafts these items at a rate far below that of our proficient contractors, e.g. on average our contractors craft 5.44 crafting tables in 60 minutes of play versus 0.19 for the foundation model. The model also crafts a non-negligible amount of wooden sticks, which are required to make wooden tools; collects various flowers and crafts dyes from them; kills zombies that appear during the night; hunts wild animals; collects various berries and mushrooms and eats them; and finds game-generated villages from which to collect various rare items from chests. The model also learned to navigate uneven terrain, swim, and pillar jump, which involves the agent repeatedly jumping and quickly placing a block below itself such that it climbs upward by making a pillar.[iv]

While training and validation loss decrease healthily over training (Fig. 4, left), loss on our contractor dataset (which the VPT model does not train on) begins increasing after 7 epochs. Contractor data could be out-of-distribution because our contractors may have a different distribution of play or because there is some impactful visual domain shift compared to videos from the web, and we provide some evidence for this phenomenon in Appendix H. While one could have expected this would be predictive of declining evaluation performance, we do not see notable game statistics from the VPT foundation model rollouts (Figure 4, right) decrease over training, and in the next section we show that BC fine-tuning performance continually improves as the VPT foundation model trains.

### 4.3 Fine-Tuning with Behavioral Cloning

Foundation models are designed to have a broad behavior profile and be generally capable across a wide variety of tasks. To incorporate new knowledge or allow them to specialize on a narrower task distribution, it is common practice to fine-tune these models to smaller, more specific datasets.[1] The VPT foundation model trained on the broad `web_clean` dataset had nontrivial zero-shot performance; it was able to craft a crafting table yet unable to go past this in the technology tree. As a case study into BC fine-tuning, we attempt to improve the VPT foundation model's ability to collect and craft these "early game" items by fine-tuning to two narrower datasets targeted at Minecraft behavior within the first few minutes of players starting in a fresh world. In the first dataset, `contractor_house`, contractors have 10 minutes to build a basic house from scratch using primarily wood, sand, and dirt. Collecting contractor data can be difficult and expensive, so we also construct a dataset `earlygame_keyword` by searching for videos online with descriptions that match keywords such as "new world", "let's play episode 1", etc.; this is a subset of `web_clean` and is labeled with the IDM. See Appendix B.4 and A.3 for full descriptions of both datasets.

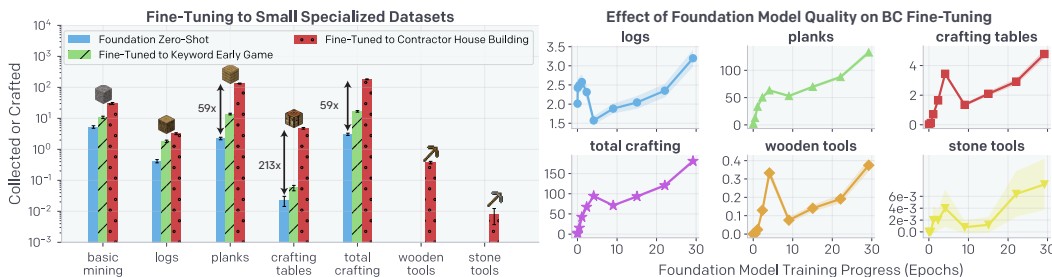

Figure 5: **(Left)** Collection and crafting rates for three policies: the zero-shot VPT foundation model, and the VPT foundation model BC fine-tuned to the `earlygame_keyword` or `contractor_house` datasets. BC fine-tuning to either dataset improves performance, including (for the `contractor_house` dataset) yielding wooden and stone tools. Proficient Minecraft players take a median of 1.2 minutes (1390 actions) to construct wooden tools and 2.3 minutes (2790 actions) to construct stone tools. **(Right)** Collection and crafting rates for VPT foundation model snapshots throughout training *after* they are BC fine-tuned to the `contractor_house` dataset. In general, crafting-related behaviors increase throughout foundation model training. Fig. 4 defines the other task terms (logs, planks, crafting tables, and total crafting).

---

[iv]Sample videos: https://www.youtube.com/playlist?list=PLNAOIb_agjf3U3rSvG_BCWqJ869NdBhcP

Fine-tuning to `earlygame_keyword` results in a large boost compared to the zero-shot foundation model: 2.5x more crafting tables, 6.1x more planks, 4.3x more logs, and 5.5x more crafting overall (Fig. 5). However, when fine-tuning to this dataset we did not see any new behaviors emerge, only a refinement of existing skills. We saw an even bigger improvement when fine-tuning to the `contractor_house` dataset: 213x more crafting tables, 59x more wooden planks, 7x more logs, and 59x more crafting over all. In addition, we saw the emergence of crafting wooden tools, which requires placing a crafting table on the ground, opening it to reveal a new crafting interface, and then using it to craft wooden tools. This entire sequence takes a proficient human player a median of 1.2 minutes (1390 consecutive actions) to accomplish. The model goes further and collects cobblestone, which requires a wooden pickaxe to mine, and crafts stone tools, requiring it to again use a crafting table; this takes a proficient human player a median of 2.3 minutes (2790 consecutive actions). We also saw this model more frequently raiding villages that randomly spawn in the game, hunting animals for food, in addition to many behaviors we saw performed by the foundation model.[v]

Despite the foundation model's zero-shot rollout performance plateauing 1/3 into training (Fig. 4, right), fine-tuning performance *does* continue to increase throughout foundation model training (Fig. 5, right). Additionally, there is a stark difference in performance when training from scratch vs. fine-tuning from the VPT foundation model (Fig. 5 right, comparing the left and rightmost points).

### 4.4 Fine-Tuning with Reinforcement Learning

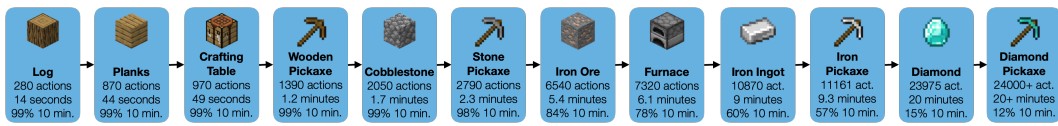

Figure 6: Typical sequence of items for obtaining a diamond pickaxe. Below each item is the median time and number of actions contractors required to obtain that item and the percentage of contractors that got the item within 10 minutes. The median time to obtain a diamond pickaxe is unknown (except that it is $> 20m$) because contractors obtained this item in less than $50\%$ of 20-minute episodes.

To demonstrate the efficacy of RL fine-tuning, we chose the challenging goal of obtaining a diamond pickaxe within 10 minutes starting from a fresh Minecraft survival world. Doing so involves acquiring a sequence of difficult-to-obtain items that require complex skills like mining, inventory management, crafting with and without a crafting table, tool use, operating a furnace, and mining at the lowest depths, where many hazards like enemies and lava exist (Fig. 6). Adding to the difficulty, progress can be easily lost by dropping items, destroying items, or dying. Obtaining a diamond pickaxe more often than not takes a proficient human over 20 minutes (24,000 actions).

Agents are rewarded for each item obtained in the sequence, with lower rewards for items that have to be collected in bulk and higher rewards for items near the end of the sequence. Agents are optimized with the phasic policy gradient[65] RL algorithm for $\sim$1.3 million episodes (roughly $1.4 \times 10^{10}$ frames). Episodes last for 10 minutes. See Appendix G.1 for reward function and RL training details. Due to computational constraints, RL experiments use a $\sim 248$ million parameter VPT model (Appendix H).

A major problem when fine-tuning with RL is catastrophic forgetting[66,67] because previously learned skills can be lost before their value is realized. For instance, while our VPT foundation model never exhibits the entire sequence of behaviors required to smelt iron zero-shot, it *did* train on examples of players smelting with furnaces. It therefore may have some latent ability to smelt iron once the many prerequisites to do so have been performed. To combat the catastrophic forgetting of latent skills such that they can continually improve exploration throughout RL fine-tuning, we add an auxiliary Kullback-Leibler (KL) divergence loss between the RL model and the frozen pretrained policy.[11]

Training from a randomly initialized policy fails to achieve almost *any* reward, underscoring how hard an exploration challenge the diamond pickaxe task is for RL in the native human action space (Fig. 7a). The model never learns to reliably collect logs, typically the first of many steps to obtaining a diamond pickaxe (Fig. 7b). RL fine-tuning from the VPT foundation model does substantially better (Fig. 7a), learning everything up to mining iron ore and crafting furnaces. (Fig. 7c). However, this agent fails at smelting an iron ingot, the next item required to get further into the tech tree, likely

[v]Sample Videos: https://www.youtube.com/playlist?list=PLNAOIb_agjf2yDSs4AqcoyPv4z_eWUiKm

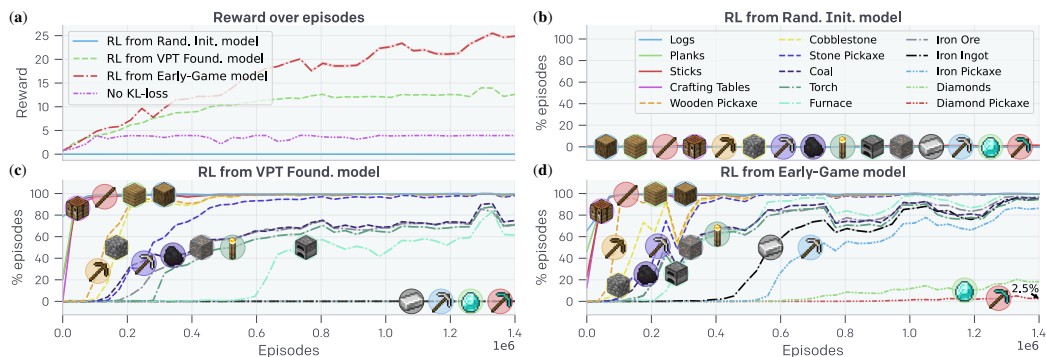

Figure 7: RL Fine-tuning results. **(a)** RL from a randomly initialized model fails to get almost any reward, RL fine-tuning from the VPT foundation model performs substantially better with a reward near 13, and RL fine-tuning from the early-game model performs best with a reward of 25. When training the early-game model without a KL loss to the original policy (*No KL-loss*) progress stalls after 100,000 episodes, suggesting that the skills necessary to make further progress have been catastrophically forgotten. **(b)** RL from a randomly initialized model occasionally collects sticks by breaking leaves (an easy but inefficient method of getting sticks that does not require logs or planks) and never learns to reliably collect logs. **(c)** RL fine-tuning from the VPT Foundation model learns everything in the curriculum up to iron ore and making furnaces, but fails to learn to use the furnace to smelt iron ingots. **(d)** RL fine-tuning from the early-game model learns to obtain (at human-level) all items in the sequence towards a diamond pickaxe and crafts a diamond pickaxe in 2.5% of episodes.

because the zero-shot probability that the VPT foundation model smelts an iron ingot is too low, even when given the prerequisite materials.

Results further improve by first BC fine-tuning the VPT Foundation Model to the `earlygame_keyword` dataset (the *early-game model*, Sec. 4.3) and then fine-tuning with RL (Fig. 7a), which in preliminary experiments we found to perform better than first fine-tuning to `contractor_house` followed by fine-tuning with RL (Appendix G.2). The three-phase training (pretraining, BC fine-tuning, and then RL fine-tuning) succeeds in learning extremely difficult tasks: it achieves over 80% reliability on iron pickaxes, almost 20% reliability on collecting diamonds, and 2.5% reliability on obtaining a diamond pickaxe (Fig. 7d). For comparison, human players given the objective of obtaining a diamond pickaxe collect these items in 57%, 15%, and 12% of episodes, respectively, meaning our model is human-level for crafting iron pickaxes and mining diamonds. Others have managed to obtain diamonds with $\sim 0.1\%$ reliability in 15 minutes [33,34] but always with a simplified action space designed to ease exploration. To the best of our knowledge, **we are the first to report non-zero success rates on crafting a diamond pickaxe**. Qualitatively, the model developed useful skills for diamond mining, such as efficient mining patterns, cave exploration, returning to previously placed objects like crafting tables, and advanced techniques like using wooden pickaxes as fuel when moving on to iron tools.[vi]

Finally, we validated the importance of the KL loss to the pretrained model during RL fine-tuning. The treatment without a KL loss obtains only items early in the sequence (logs, planks, sticks, and crafting tables) limiting its reward (Fig. 7a). This failure to progress further into the sequence is likely because, while the initial skills of chopping logs and crafting planks are being learned with RL, subsequent skills like crafting a wooden pickaxe are lost due to catastrophic forgetting.

## 4.5 Data Scaling Properties of the Foundation Model

In this section we validate a core hypothesis behind this work: that it is far more effective to use labeled contractor data to train an IDM within the VPT method than it is to directly train a BC foundation model from that same small contractor dataset. If we could cheaply collect a labeled contractor dataset of a similar order of magnitude as `web_clean`, then this would not be important; however, collecting that scale of data would have cost millions of dollars. Figure 8 compares

---

[vi]Videos found at https://www.youtube.com/playlist?list=PLNAOIb_agjf3e_UKweM5pQUSfTw8r-Wfc

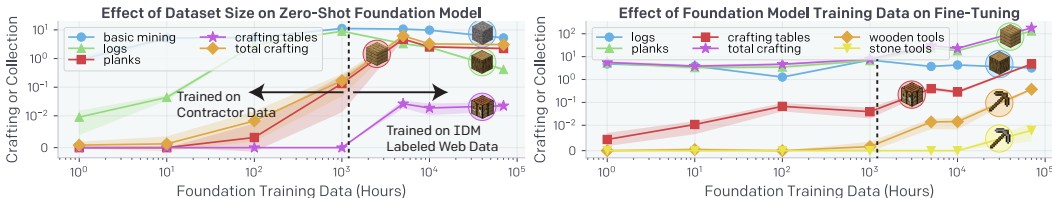

Figure 8: **(Left)** Zero-shot rollout performance of foundation models trained on varying amounts of data. Models to the left of the dashed black line (points ≤1k hours) were trained on contractor data (ground-truth labels), and models to the right were trained on IDM pseudo-labeled subsets of `web_clean`. Due to compute limitations, this analysis was performed with smaller (71 million parameter) models except for the final point, which is the 0.5 billion parameter VPT foundation model. **(Right)** The corresponding performance of each model *after* BC fine-tuning each model to the `contractor_house` dataset.

foundation models trained on increasing orders of magnitude of data from 1 hour up to the full ∼70k `web_clean` dataset. Foundation models trained up to and including 1k hours are trained on the IDM contractor data, and those trained on 5k hours and above are trained on subsets of `web_clean`, which does not contain any IDM contractor data. Scaling training data increases log collection, mining, and crafting capabilities. The zero-shot model only begins to start crafting crafting tables at over 5000 hours of training data. When fine-tuning each foundation model to `contractor_house`, we see that crafting rates for crafting tables and wooden tools increase by orders of magnitude when using the entire ∼70k hour `web_clean` dataset. We furthermore only see the emergence of crafting stone tools at the largest data scale.

## 4.6 Effect of Inverse Dynamics Model Quality on Behavioral Cloning

This section investigates how downstream BC performance is affected by IDM quality. We train IDMs on increasingly larger datasets and use each to independently label the `earlygame_keyword` dataset (this smaller dataset was chosen due to a limited compute budget). We then train a BC model from scratch on each dataset and report game statistics for each model as a function of IDM contractor dataset size (Fig. 9).

IDMs trained on at least 10 hours of data are required for any crafting, and the crafting rate increases quickly up until 100 hours of data,

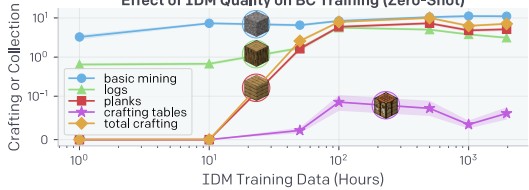

Figure 9: Zero-shot performance of BC models trained from scratch on the `earlygame_keyword` dataset labeled with IDMs that were trained on increasing amounts of contractor data.

after which there are few to no gains and differences are likely due to noise. Similarly, crafting tables are only crafted after 50 or more hours of IDM data, and again gains plateau after 100 hours. While in all previous experiments we use our best IDM trained on 1962 hours of data, these results suggest we could reduce that number to as low as 100 hours.

## 5 Discussion and Conclusion

The results presented in this paper help pave the path to utilizing the wealth of unlabeled data on the web for many sequential decision domains. Compared to representation learning methods, e.g. generative video modeling, VPT offers the exciting possibility of directly *learning to act* during pretraining and using these learned behavioral priors as extremely effective exploration priors for RL. VPT could even be an effective representation learning method for downstream tasks that do not require acting, e.g. video captioning, because arguably the most important information in any given scene would be present in features trained to correctly predict the distribution over future human actions. We leave this intriguing direction to future work.

Future work could improve results with more data (we estimate we could collect >1M hours) and larger, better-tuned models. Our internet data was fairly noisy and varied (players choose their own graphics settings); we hope future work will investigate even noisier sources of data, as well as how to use both first and third person demonstrations. Furthermore, all models in this work condition on past observations only; we cannot ask the model to perform specific tasks. Appendix I presents preliminary experiments on conditioning our models on closed captions (text transcripts of speech in videos), showing they become weakly steerable; we believe this a rich direction for future research. By definition behavioral priors must predict actions, and in this work we found this objective sufficient to train capable agents; however, a fruitful direction could be incorporating auxiliary representation learning objectives (e.g. contrastive losses, environment dynamics modeling, etc.) to reduce the sample complexity of both the IDM and foundation models. Similarly, it would be interesting to see if VPT could benefit from pretraining its attention layers with a language modeling task as in Li et al. [68] and Reid et al. [69] Loss was not consistently correlated with downstream evaluation metrics (Sec. 4.2), which often made progress slow. Another worthwhile future direction would be to investigate the correlation between various training metrics and downstream evaluations.

For RL fine-tuning we only experimented with a standard policy gradient based RL algorithm (PPG); an interesting future direction would be to investigate how well VPT can be combined with other RL algorithms, e.g. off-policy or model based. Furthermore, we showed the efficacy of fine-tuning VPT with RL using a very difficult, albeit handcrafted, reward function aimed at crafting diamond tools. We hope future work will combine VPT with methods that can generate more generic reward functions, e.g. natural language based reward functions as proposed in MineDojo [70] (released after this paper). Finally, while we do not anticipate any direct negative societal impacts from the models trained in this work, as VPT improves and expands to other domains it will be important to assess and mitigate harms that emerge with other forms of pretraining on internet datasets, such as emulating inappropriate behavior. [71]

In conclusion, VPT extends the paradigm of training large and general purpose behavioral priors to sequential decision domains that have commonly available unlabeled internet data. Our models exhibited impressive zero-shot behavior and, when fine-tuned with RL, achieved an unprecedented result of crafting a diamond pickaxe in Minecraft (all the more difficult given the human interface). We further showed that contractor data is far better used within the VPT pipeline than to train a foundation model directly and that only a small amount of contractor data (about $2000 USD) was required to unlock massive amounts of unlabeled online data for use in BC. Finally, learning with the human keyboard and mouse interface is highly general and allows losslessly modeling the entire distribution of human behavior. While we only experiment in Minecraft, we believe that VPT provides a general recipe for training behavioral priors in hard, yet generic, action spaces in any domain that has a large amount of freely available unlabeled data, such as computer usage.

## Acknowledgements

We thank the following people for helpful discussions and support: Bob McGrew, Ken Stanley, Joel Lehman, Ilya Sutskever, Wojciech Zaremba, Ingmar Kanitscheider, David Farhi, Glenn Powell, Jonathan Gordon, and the OpenAI supercomputing team, especially Christian Gibson, Ben Chess, and Christopher Berner.

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
