# Supplementary Information

## A    Collecting Internet Data

### A.1    Initial Unclean Dataset Curation

Our goal was to curate a video dataset of Minecraft gameplay from the survival game mode. Additionally, we prefer the data come from game modes as close as possible to our evaluation environment, meaning preferably coming from Minecraft version 1.16, being on a computer (which uses a mouse and keyboard vs. video game controllers with keypads and other buttons), being single- (vs. multi-) player, and having the default look of the game (vs. modifications that alter that style, such as to make it look realistic). To try to accomplish these goals, we collect a dataset by performing keyword searches of publicly available videos on the internet. A list of search queries we used are given in Table 1.

| minecraft survival longplay |
| --- |
| minecraft gameplay no webcam |
| minecraft gameplay survival mode |
| minecraft survival tutorial |
| minecraft survival guide |
| minecraft survival let's play |
| minecraft survival for beginners |
| minecraft beginners guide |
| ultimate minecraft starter guide |
| minecraft survival guide 1.16 |
| minecraft how to start a new survival world |
| minecraft survival fresh start |
| minecraft survival let's play episode 1 |
| let's play minecraft episode 1 |
| minecraft survival 101 |
| minecraft survival learning to play |
| how to play minecraft survival |
| how to play minecraft |
| minecraft survival basic |
| minecraft survival for noobs |
| minecraft survival for dummies |
| how to play minecraft for beginners |
| minecraft survival tutorial series |
| minecraft survival new world |
| minecraft survival a new beginning |
| minecraft survival episodio 1 |
| minecraft survival епизод 1 |
| minecraft survival 1. bölüm |
| i made a new minecraft survival world |

Table 1: Search terms used for generating the initial web dataset.

For videos that have metadata available, we perform an additional step of metadata-based filtering to eliminate videos that do not fit our target distribution. In this step, we look for a list of blacklist keywords in the video title and description and reject videos that contain these terms. The blacklist keywords we use are: {ps3, ps4, ps5, xbox 360, playstation, timelapse, multiplayer, minecraft pe, pocket edition, skyblock, realistic minecraft, how to install, how to download, realmcraft, animation}. This process yielded us ∼270k hours of unlabeled data, which we filter down to only a "clean" subset as described in the next section.

### A.2    Training a Model to Filter out Unclean Video Segments

We restrict the scope of this work to the Minecraft Survival game mode and therefore limit our training dataset to clips that are obtained from this mode that are relatively free from visual artifacts.

To do so, we asked contractors to label a set of random video frames (images) from Minecraft videos (N=8800). These images were from a random subset of the videos we collected toward the beginning of the project (Section A.1).

### A.2.1 Label Collection

We asked 5 workers on Amazon Mechanical Turk (mTurk) that we selected with a sample qualification task to label random screen capture images to be used in training the classifier. A sample worker interface that the workers saw on mTurk is given in Figure 10.

We asked workers to label videos as being in one of the following three categories (see Figure 11 for visual examples of each class):

1. `Minecraft Survival Mode - No Artifacts`: Video frames (images) that correspond to the Minecraft Survival game mode that do not contain any non-game visual artifacts (e.g. subscribe buttons, channel logos, advertisements, picture-in-picture of the narrator, etc.).

2. `Minecraft Survival Mode - with Artifacts`: Video frames (images) of the Minecraft Survival game mode that include such visual artifacts.

3. `None of the Above`: Video frames (images) that are not from the Minecraft survival game mode, including those from other Minecraft game modes such as *creative* mode or even other games/topics entirely.

The full set of instructions workers received are as follows (note that we also included multiple image examples from each category in the worker instructions, similar to the sample subset provided in Figure 11):

---

Please help us identify screenshots that belong only to the survival mode in Minecraft. Everything else (Minecraft creative mode, other games, music videos, etc.) should be marked as `None of the above`. Survival mode is identified by the info at the bottom of the screen:

- a health bar (row of hearts)
- a hunger bar (row of chicken drumsticks)
- a bar showing items held

**Survival Mode**
Valid survival mode videos have health/hunger bars and an item hotbar at the bottom of the screen.
**Creative Mode**
Creative mode only has an item hotbar and should be classified as `None of the Above`.

**Label Descriptions**

- `Minecraft Survival Mode - No Artifacts`: These images will be clean screenshots from the Minecraft survival mode gameplay without any noticeable artifacts.

- `Minecraft Survival Mode - with Artifacts`: These images will be valid survival mode screenshots, but with some added artifacts. Typical artifacts may include image overlays (a logo/brand), text annotations, a picture-in-picture of the player, etc.

- `None of the Above`: Use this category when the image is not a valid Minecraft survival screenshot. It may be a non-Minecraft frame or from a different game mode. In non-survival game modes such as the creative mode, the health/hunger bars will be missing from the image, the item hotbar may or may not be still present.

---

In total, we spent $319.96 on human labeling experiments on mTurk, of which $159.98 was directly paid to workers. The remaining amount was spent towards Amazon platform fees. The workers received $0.01 per labeled image, at an hourly compensation of $7.20 (based on an estimated labeling time of 5 seconds/image – in our internal sample run of the same task, we found the average labeling time to be < 3 seconds).

Since we perform rigorous keyword and metadata based filtering of videos (as described in A.1) from which we served sample images to be labeled, serving offensive content to workers was extremely

low risk and no such images were detected during our manual checks. We only collected labels during our experiment, and the workers were fully anonymized via the mTurk platform, therefore no personally identifiable information (PII) was collected.

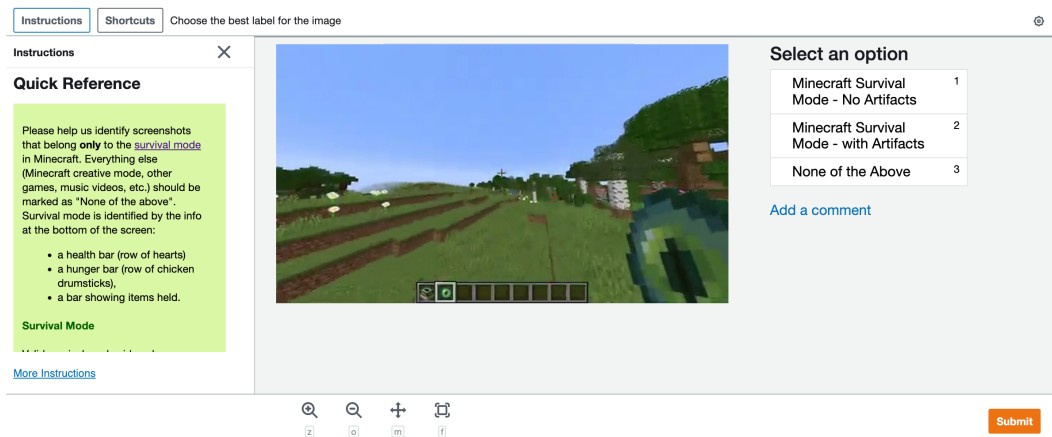

Figure 10: Amazon Mechanical Turk worker interface showing an example labeling task

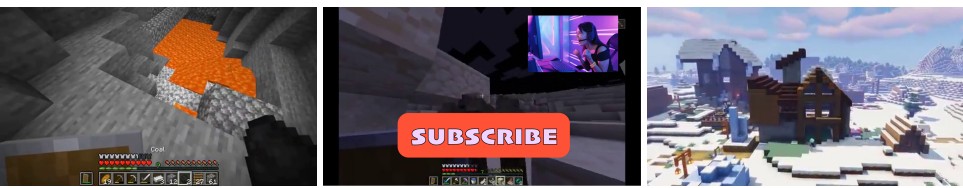

Figure 11: **(Left)** Sample image for Class 1: `Minecraft Survival Mode - No Artifacts`. **(Middle)** Sample image for Class 2: `Minecraft Survival Mode - with Artifacts` – Image contains annotations and picture-in-picture of the narrator. **(Right)** Sample image for Class 3: `None of the Above` – Image is missing the hotbar as well as health and armor bars, indicating that it was not captured during survival mode gameplay

### A.2.2 SVM Training

With the image labels collected as described in the previous section, we trained a classifier to extract video segments that consist of frames from the `Minecraft Survival Mode - No Artifacts` category. Given a set of labeled images, we obtain embeddings for each image using the RN50x64 ResNet CLIP Model. [6] This is a ResNet-based CLIP model that is scaled up to have approximately 64x the compute of a ResNet-50. We then train a Support Vector Machine (SVM) using the RBF kernel to obtain a frame classifier. We use the Scikit-learn [72] SVM implementation with the parameter configuration given in Table 2.

Finally, we apply the classifier to frames of raw video sequences at a rate of 3 frames/second. We filter for videos that consist of at least 80% "clean" frames at this stage (Classes `Minecraft Survival Mode - with Artifacts` and `None of the Above` are both considered not clean). From this set, we apply a median filter (with a kernel size of 7) to the labels and segment videos by splitting the "clean" segments that are at least 5s in duration. The result of this is our final `web_clean` dataset.

### A.2.3 Effect of Data Cleaning

To validate the data cleaning SVM, we first construct a dataset which is comprised of video clips of the first 5 minutes of any video with the same early game keywords used in the rest of the paper but without using the SVM. This dataset has 14785 hours of video. Because our `early_game` dataset is comprised of clean clips that start within the first 5 minutes of the internet video, we create a new dataset in which we only take clean clips that go up to the 5 minute mark and cut them there such that

| CLIP Model Specification | RN50x64 (see text) | |
|---|---|---|
| CLIP Input Image Resolution | 448x448x3 | |
| CLIP Embedding Feature Length | 1024 | |
| SVM Parameters | Kernel | `rbf` |
| | C | 20 |
| | Gamma | `scale` |
| Sample Size | Class 1 | 2200 |
| | Class 2 | 2200 |
| | Class 3 | 4400 |

Table 2: Feature Extraction Details and SVM Configuration. The parameters are for the SVM implementation in Scikit-learn [72].

these two datasets are a more fair comparison. This new early game dataset has 2094 hours. We train a 71M parameter VPT model from scratch (no fine-tuning) on each dataset. For a fair comparison, we train the cleaned dataset for 20 epochs and the uncleaned dataset for 2.8 epochs such that each model has the same compute budget. We show the crafting and collection rates for each model at the end of training in Figure 12 and find that the data data cleaning SVM is quite important in eliciting the more complex crafting behaviors. We see ∼10 times the crafting rate of crafting tables for the model trained on clean data, and the model trained on cleaned data is able to craft some wooden tools whereas the model trained on all data is not.

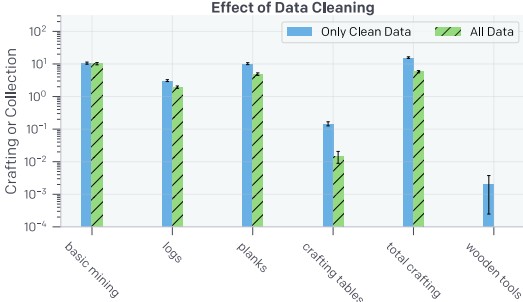

Figure 12: Effect of Data Cleaning. We compare two models, one trained on all early game data and one trained on clean early game data filtered with our SVM.

## A.3  `early_game` Dataset

The `early_game` dataset is a ∼3000 hour subset of `web_clean` targeted at "early game" Minecraft behavior, i.e. instances where players start in a fresh world with no items. We obtain the metadata text that accompanies the videos in `web_clean` and determine whether any of the following regular expressions match:

- (ep|episode|eps|day|session|sesh|chapter|chap .|series|part|parte|pt|round|day|tâp|bölüm|episodio|эпизод|эпизод)( )*(\.1|#1|1|\.01|#01|01|one[^0-9]|$)

- start

- beginning

- (new|fresh|clean).*(world|game|play)

- from scratch

From this set of videos, we take only video snippets that began within the first 5 minutes of each video.

## B  Contractor Data

### B.1  Recording Contractor Play

Our contractors use a custom Minecraft recorder that we built that records their actions and game video feeds as they play. The recorder is implemented using the MCP-Reborn (github.com/Hexeption/MCP-Reborn) modding package. To ensure that the recorder environment is as close as possible to the Minecraft environment used for RL rollouts and evaluations (Appendix C), we use the same underlying game engine for both. The recorder is a Java app that runs in a window mode, with constant resolution of 1280x760. Brightness is set to 0 (the "gloomy" setting in Minecraft), which is the default setting. Other graphics settings (field of view, GUI scale) are fixed to the values used in the Minecraft environment (C.1); we explicitly prevented users from changing graphics settings. Unlike the environment, the recorder allows all keyboard key presses and continuous (as opposed to binned) mouse actions. On every game step (or "tick") the frame buffer used to display the game window is downsized to 640x360 and written into a video file. In-game actions are recorded in a separate JSONL file (a text file where each line is a JSON-formatted string). All recordings are chunked into 5 minute clips: after each 5 minute segment of contractor game play the recorder automatically uploads the video file, the JSONL file with actions, as well as a Minecraft state file. To ensure that contractors cannot corrupt each other's data, we provided every contractor with an individual cloud bucket, as well as with credentials giving write access only to that bucket. Credentials also included adjective-adjective-noun names (e.g. grumpy-amethyst-chipmunk), generated with the `namegenerator` python package to ensure contractor anonymity when we publish the data.

### B.2  Contractor Contract

We recruited contractors by posting the following offer on the UpWork freelancing platform.

> "We are collecting data for training AI models in Minecraft. You'll need to install java, download the modified version of Minecraft (that collects and uploads your play data), and play Minecraft survival mode! Paid per hour of gameplay. Prior experience in Minecraft not necessary. We do not collect any data that is unrelated to Minecraft from your computer."

We had the applications open for a day, and then randomly selected 10 applicants for the first round of contractors. Later in the project, as we needed more data and as some contractors asked to terminate their contracts, we added more applicants from the original pool as well as referrals from the currently working contractors. The contractors were paid $20 per hour (minus Upwork platform fees and applicable taxes). All of the results presented in this paper are based on about 4,500 hours of data (including data recorded to gather statistics of human play that was not used for training), which cost us around $90,000. Over the course of the project, we collected some data we did not use due to bugs in the recorder and for some ideas we ultimately did not pursue. In total, we spent about $160k for contractor compensation over the course of the project. However, as we discuss in Sec. 4.6, we could likely obtain most of our results with an IDM trained using only $2000 worth of data, i.e. the foundation VPT model, BC fine-tuning to the `earlygame_keyword` dataset, and the RL fine-tuning results. Collecting the `contractor_house` dataset cost about $8000. Because we used the IDM trained on about 2000 hours of contractor data, the actual cost of contractor data for those results was around $40,000.

In early stages of the project, we were planning to use contractor data solely for the purpose of training the IDM. As such, no specific tasks were given, other than "play the survival mode of Minecraft like you normally would." Later in the project, we requested that contractors perform specific tasks in Minecraft, such as:

- Collect as many units of wood as possible, using only wooden or stone tools (`treechop`)
- Start a new world every 30 minutes of game play
- Build a basic house in 10 minutes using only dirt, wood, sand, and either wooden or stone tools (`contractor_house`, more details below in Appendix B.4).
- Starting from a new world and an empty inventory, find resources and craft a diamond pickaxe in 20 minutes (`obtain_diamond_pickaxe`). This dataset was used to obtain

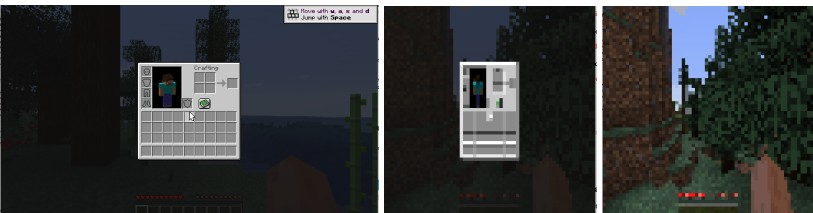

Figure 13: **(Left)** Sample of a Minecraft frame in the original resolution (640x360) with an in-game GUI open. The mouse cursor can be seen in the center of the image. This particular GUI shows the player's inventory and can be used to craft very basic items. **(Middle)** We downsample images to 128x128 for computational reasons. Shown is a downsampled observation with an in-game GUI for crafting. This is the resolution used by our models. **(Right)** A 128x128 observation as seen by our models without in-game GUI. The health, hunger, hotbar overlays, and agent hand can be seen in the lower part of the image.

statistics for how long it takes humans on average to complete this task (and the subtasks required to complete it) when obtaining a diamond pickaxe is their goal.

Since we only recorded in-game events and videos, the data does not include personally identifiable information. That being said, the contractors could theoretically use Minecraft's open-world property to generate personally identifiable information and/or offensive content (e.g. by using Minecraft blocks to write their name or offensive messages, then finding a spot from which the message would be visible). In practice, we have not seen any attempts to do so in the contractor videos that we watched. Of course, we train our BC models on videos from the internet of people playing Minecraft, and if such behavior is in those videos our model could also potentially learn it, although we expect such behavior is rare enough that our model would not be likely to reproduce it.

### B.3   Data for the Inverse Dynamics Model.

Since the IDM's task is to infer actions given the video, any labelled data is appropriate for IDM training. In practice, we included general gameplay as well as the `treechop` task data described in the previous section, which amounted to a total of 1962 hours. Due to collecting datasets like `contractor_house` only at late stages of the project, they were not included in IDM training.

### B.4   contractor_house.

The `contractor_house` contains about 420 hours of data. We asked contractors to build a basic house in 10 minutes, using only basic dirt, wood, and sand, blocks. Each trajectory starts in a newly generated world and a timer forcibly ends a trajectory after a 20 minute time limit. For this task, many contractors chose to begin their trajectories by crafting basic tools and building blocks, specifically it was common for the first 2 minutes to be spent crafting a wooden pickaxe and then mining stone for an assortment of stone tools before gathering more building blocks and beginning to create their structure.

## C   Minecraft environment details

Our Minecraft training environment is a hybrid between MineRL[28] and the MCP-Reborn (github.com/Hexeption/MCP-Reborn) Minecraft modding package. Unlike the regular Minecraft game, in which the server (or the "world") always runs at 20Hz and the client runs as fast as rendering can complete (typically at 60-100Hz), in our version the client and server run in the same thread at the same frequency. This allows us to run the environment slower or faster than real time, while avoiding artifacts like missing chunks of the world. The action and observation spaces are similar to those of MineRL environments and are described in more detail in the following subsections. The environment also returns diagnostic information, such as in-game stats, contents of the agent's inventory, whether any in-game GUI is open, etc., which we use for tracking and recording but not as inputs to the models.

All evaluations use the default Minecraft world generation options as well as randomly generated worlds. The episode length is 10 minutes for RL experiments and 60 minutes for BC model evaluations. The agent can "die" in a number of ways, such as staying under water for too long and drowning, being killed by hostile mobs, or falling from a tall structure. We do not terminate the episode on agent "death". Instead, just as for humans in the regular Minecraft game, the agent drops all its items when it dies and respawns at a random spot close to the initial spawning spot in the same Minecraft world. The policy state is not masked on death, so the model can remember the fact that it has died and act accordingly.

## C.1 Observation space

The environment observations are simply the raw pixels from the Minecraft game that a human would see. Unlike MineRL, we do not remove overlays like the hotbar, health indicators, and the animation of a moving hand shown in response to the attack or "use" actions. The field of view is 70 degrees, which corresponds to the Minecraft default. GUI scale (a parameter controlling the size of the in-game GUI) is set to 2, and brightness is set to 2 (which is not a Minecraft default, but is very frequently used in online videos). The rendering resolution is 640x360, which is downsampled to 128x128 before being input to the models. We empirically found 128x128 to be the smallest resolution for which in-game GUI elements are still discernible, and then chose that to minimize compute costs. Whenever an in-game GUI is open, we additionally render an image of a mouse cursor at the appropriate mouse position to match what a human player's operating system does (Fig. 13).

## C.2 Action space

Our action space includes almost all actions directly available to human players, such as keypresses, mouse movements, and clicks. The specific binary actions we include are shown in Table 3.

One difference between the human action space and our agent's is that we disallow typing arbitrary letters, which is only useful for entering text into the search bar of the crafting recipe book. Humans can either do that or browse the recipe book with the mouse, the latter of which our agent can still do. However, because we do allow the agent to press letters that are also shortcuts for actions (e.g. outside of the GUI, the "W" key triggers the `forward` action) agents are able to press a few keys within the GUI (W, A, S, D, E, Q) that produce letters if the recipe book search bar is selected. We have not seen agents attempt to search the recipe book with these letters. Instead, our agents navigate the recipe book with the mouse or craft by dragging items around the crafting window.

In addition to the binary (on/off) keypress actions, our action space also includes mouse movements. As with human gameplay, when in-game GUIs are not open, mouse X and Y actions change the agent's yaw and pitch, respectively. When a GUI is open, camera actions move the mouse cursor. Mouse movements are relative (i.e. they move the mouse or camera relative to the current position, and thus their effect depends on the current position).

Inventory interaction in Minecraft requires fine-grained mouse movements to achieve tasks such as crafting and smelting, while mining and navigating the world can be achieved with coarser mouse action. To be able to achieve both with the same action space, we implemented mouse movements as a set of discrete actions with foveated binning along each axis (Fig. 14), which in preliminary experiments we found to improve crafting performance.

We use $\mu$-law quantization to achieve foveated binning

$$F(x) = \text{sgn}(x) \frac{\log\left(1 + \mu \left|\frac{x}{V_{\max}}\right|\right)}{\log(1 + \mu)} V_{\max} \tag{2}$$

where $x$ is the raw camera pixel value along each axis, $F(x)$ is the transformed value in the foveated action space, sgn is the sign function, $V_{\max}$ is the maximum camera value beyond which values are clipped, and $\mu$ is the parameter that determines the slope of foveation. In our experiments, we used $V_{\max} = 10$ and $\mu = 10$ followed by a quantization step that assumes 11 bins for each axis.

| Action | Human action | Description |
|---|---|---|
| forward | W key | Move forward. |
| back | S key | Move backward. |
| left | A key | Strafe left. |
| right | D key | Strafe right. |
| jump | space key | Jump. |
| inventory | E key | Open or close inventory and the 2x2 crafting grid. |
| sneak | shift key | Move carefully in current direction of motion. In the GUI it acts as a modifier key: when used with `attack` it moves item from/to the inventory to/from the hotbar, and when used with `craft` it crafts the maximum number of items possible instead of just 1. |
| sprint | ctrl key | Move fast in the current direction of motion. |
| attack | left mouse button | Attack; In GUI, pick up the stack of items or place the stack of items in a GUI cell; when used as a double click (attack - no attack - attack sequence), collect all items of the same kind present in inventory as a single stack. |
| use | right mouse button | Place the item currently held or use the block the player is looking at. In GUI, pick up the stack of items or place a single item from a stack held by mouse. |
| drop | Q key | Drop a single item from the stack of items the player is currently holding. If the player presses ctrl-Q then it drops the entire stack. In the GUI, the same thing happens except to the item the mouse is hovering over. |
| hotbar.[1-9] | keys 1 − 9 | Switch active item to the one in a given hotbar cell. |

Table 3: Binary actions included in the action space. `https://minecraft.fandom.com/wiki/Controls` has more detailed descriptions of each action.

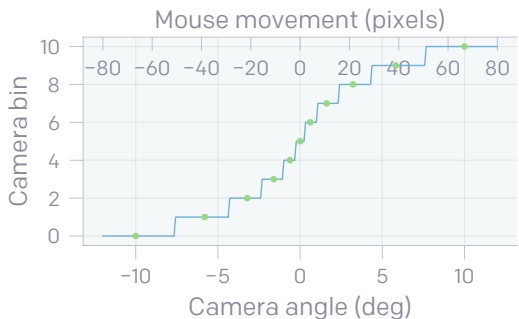

Figure 14: Relative camera angle or mouse movement in pixels vs. action bin. The same binning is used for both X and Y coordinates. The binning is foveated, meaning that binning is more fine-grained for smaller movements and more coarse-grained for larger movements. There are 11 bins for each axis (X and Y). The center of each bin (indicated with green circles) is used when un-discretizing movements (that is, when converting from an action expressed as a bin to a camera angle or mouse movement).

# D   Inverse Dynamics Model Training Details

## D.1   IDM Architecture

The IDM model has approximately 0.5 billion trainable weights. The input to the IDM is 128 consecutive image frames (128 frames of video), each of which has dimensions $128 \times 128 \times 3$. The IDM is tasked with predicting the action at each frame. All image pixel values are first divided by 255.0 such that they lie within the range $[0, 1]$. The first layer of the IDM is a 3-D convolution with 128 learnable filters with a temporal kernel width of 5 and spatial kernel widths of 1. This convolution is non-causal, meaning that embeddings at time index $t$ are functions of pixel values at times $t - 2$, $t - 1$, $t$, $t + 1$, and $t + 2$. We found this layer to be extremely important in IDM training as it incorporates neighboring temporal information immediately, and we show results comparing IDM performance with and without this layer in Figure 15. This comparison was made on the default (1962-hour) IDM dataset.

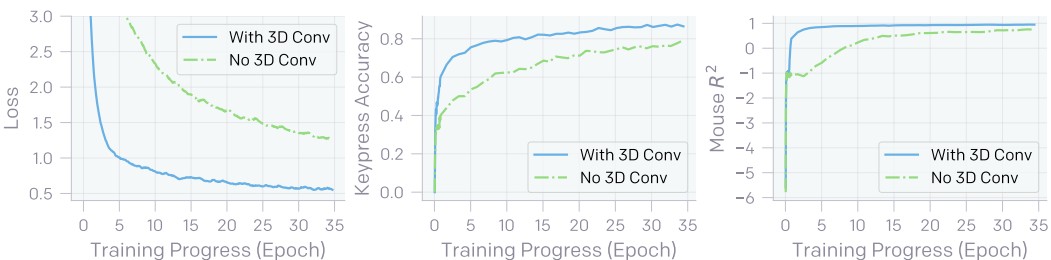

Figure 15: Effect of 3-D Convolution in the IDM Architecture.

This initial temporal convolutional layer is followed by a ResNet[63] image processing network. In this part of the model, no extra temporal information is shared between neighboring frames; however, since each frame was first processed with the temporal convolution, some temporal information is present at this stage. The ResNet image processing network is comprised of three subsequent stacks with widths $W = \{64, 128, 128\}$. Each stack is comprised of, in order, (1) an initial 3x3 convolutional layer with 1-pixel zero padding at the embedding boundary (such that the outgoing embedding dimensions are the same as the incoming embedding dimension) with $W$ output channels, (2) a 3x3 max pooling with stride 2 and padding 1 such that the embedding width and height are halved, and (3) two classic ResNet blocks as defined in He et al.[63] with each layer also having $W$ output channels.

The output of the ResNet stack is flattened into a 1-dimensional vector of size $2^{17} = 131072$ (one vector for each frame in the video) such that at this stage there are 128 vectors of size 131072. Each vector is independently processed with two frame-wise dense layers with 256 output activations and then 4096 output activations, respectively. The result is then fed through 4 subsequent non-causal (umasked) residual transformer[73] blocks. Each block first has an unmasked attention layer, i.e. frames may attend to future frames, with 32 attention heads of dimension 128 each and a surrounding residual connection that skips this layer. The embedding is then passed through a frame-wise dense layer with output dimension 16384 and another with output dimension returning to 4096; a single residual connection skips past this pair of frame-wise dense layers (not skipping past each layer separately, but skipping the pair). All dense layers have their weights tied through time, so each frame in the video is processed with the same weights.

Finally, independent dense layer heads for each action are pulled from the final embedding – a 2 class on/off categorical parameterized with a softmax for each available key as well as a 11-way categorical for both the discretized horizontal and vertical mouse movements (See Appendix C.2 for details on the action space).

Each dense layer or convolutional layer in the network is preceded by a layernorm[74] and followed by a ReLU non-linearity. Weights are initialized with Fan-In initialization[75] and biases are initialized to zero.

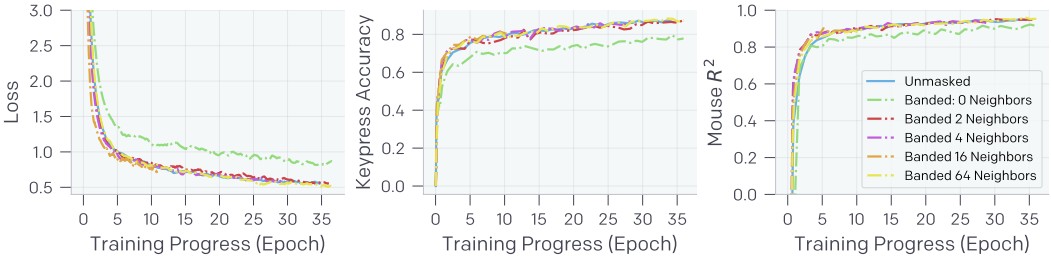

Figure 16: Effect of Attention in the IDM Architecture. Here we compare our baseline architecture where the attention layers are unmasked, i.e. they can attend to all frames in the sequence of 128 frames, to cases where the attention layers are masked to a certain number of neighbors. For example, Banded 0 Neighbors indicates that the attention layer cannot attend to any other frames and is therefore equivalent to MLP layers with residual connections. Banded 2 Neighbors indicates that each frame can attend to the previous and next 2 frames in the video sequence. The architectures all performed comparably except for Banded 0 Neighbors, which performs quite a bit worse.

## D.2   IDM Training

The total loss for the network is the sum of each independent action prediction loss (one for each key and one for both mouse directions). Each independent loss is the negative log-likelihood of the correct action. We use the ADAM[76] optimizer with a linear learning rate decay. We use an initial learning rate of 0.003, a batch size of 128 (where each item in the batch is a video sequence of 128 frames), and a weight decay of 0.01. Hyperparameters were tuned in preliminary experiments. The IDM is trained on our contractor collected dataset for 20 epochs. This took 4 days on 32 A100 GPUs.

We add data augmentation to each video segment; augmentations are randomly sampled once per segment such they are temporally consistent. Using the Pytorch[77] transforms library, we adjust the hue by a random factor between -0.2 and 0.2, saturation between 0.8 and 1.2, brightness between 0.8 and 1.2, and contrast between 0.8 and 1.2. We also randomly rotate the image between -2 and 2 degrees, scale it by a random factor between 0.98 and 1.02, shear it between -2 and 2 degrees, and translate it between -2 and 2 pixels in both the $x$ and $y$ dimensions.

Due the large computational cost of running all of the experiments in this paper, training results are from one run of training (for IDM, BC, and RL training): this non-ideal situation is mitigated because deep learning training tends to be low variance[78,79] and because we often have data points from sweeps (e.g. on dataset size) that suggest overall trends.

## D.3   Generating Pseudo Labels with the IDM

Section 4.1 shows that inverse dynamics modeling is a much easier task than behavioral cloning because IDMs can be non-causal. The IDM is trained to simultaneously predict all 128 actions for each video sequence, so the IDM will effectively be causal for frames at the end of the video clip because future frames are not included in the sequence. For this reason, we apply the IDM over a video using a sliding window with stride 64 frames and only use the pseudo-label prediction for frames 32 to 96 (the center 64 frames). By doing this, the IDM prediction at the boundary of the video clip is never used except for the first and last frames of a full video.

# E   Foundation Model Behavioral Cloning

## E.1   Foundation Model Architecture

The behavioral cloning model architecture is the same as the IDM architecture described in Appendix D.1 except that we modify the architecture so that it is causal (i.e. cannot see the future when making predictions). This means the BC architecture does not have the initial non-causal convolution the IDM has (this layer is omitted completely). Furthermore, the residual transformer layers are now causally masked (as is standard in language modeling) and we do Transformer-XL-style[80] training

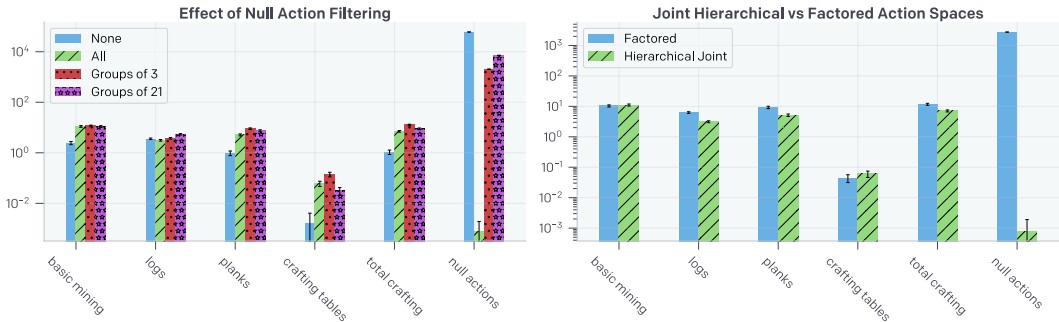

Figure 17: **(Left)** Effect of Null Action Filtering during training. We compare environment metrics and number of sampled null action during rollouts (rightmost group of columns) for the following treatments: no null action filtering (blue), filtering all null actions (green), filtering only groups of 3 or more null actions (red), and filtering only groups of 21 or more null actions (purple). **(Right)** Hierarchical versus Factored Action Spaces.

where frames can attend to keys and values from past batches within the same video. We also use a Transformer-XL-style relative attention position embedding.

## E.2 Null Action Filtering

The most common action humans take is the null action (no keypresses or mouse movements), which accounts for 35% of all actions they take. Among other reasons, a player may take the null action to wait for something in the game to finish, to pause between actions, or to take a break to grab a glass of water. Early on in the project we found that the BC model would take a much larger fraction than 35% of null actions, often upwards of 95%. In order to prevent this behavior we removed frames with null actions from the dataset. We compare a few different treatments: we filter nulls if there have been 1, 3, or 21 frames of consecutive null actions, and include a treatment that does not perform any null filtering. Null action filtering generally helps, increasing all crafting rates (Figure 17 left). Filtering only groups of 3 performed slightly better than filtering all null action or groups of 21. Initial experiments indicated that filtering all null actions was better; however, after further model tuning and after we had already trained our largest models, we found that filtering only groups of 3 or more null actions performed best. Due to compute constraints we were not able to redo all experiments with this setting, but doing so would be a reasonable choice for any future work.

## E.3 Joint Hierarchical Action Space

We originally worked with a factored action space, where each keypress could be independently on or off, and this choice was independent of whether the mouse was being moved. This could cause issues for modeling the human behavior distribution exactly. Say for a given state, humans either with 50% probability (a) move forward and attack or with 50% probability (b) move left and drop their item. The best a factored distribution can do is to assign 50% probability to each of the 4 constituent actions because it chooses to press each button simultaneously and independently. See Appendix C.2 for details on the entire action space.

For this reason, we implemented a joint distribution over actions; however, the full joint distribution over 20 binary buttons and two mouse movement dimensions discretized into 11 bins each would result in in $2^{20} \times 11^2 \approx 1.2 \times 10^8$ possible combinations. This is far too large for many reasons, e.g. the final layer from the transformer stack with a dimension of 4096 would need to be mapped to each combination resulting in $4096 \times 1.2 \times 10^8 \approx 5.2 \times 10^{11}$ parameters for this final layer alone. In order to reduce this we noted that many buttons in Minecraft have no effect when simultaneously pressed; for example, if a player tries to move forward and backward at the same time, they remain in place. Below we list the the sets of mutually exclusive actions. Furthermore, the inventory button is exclusive with all other buttons and mouse movement.

| Mutually Exclusive Actions |
|:---:|
| `forward, back` |
| `left, right` |
| `sprint, sneak` |
| `hotbar.[1-9]` |

Even reducing the joint action space to reflect these mutually exclusive combinations still results in a huge action space when combined with the discretized mouse movements, i.e. $3^3 \times 10 \times 2^4 \times 11^2 + 1 \approx 5.2 \times 10^5$. This calculation results from $3^3$ for the 3 sets of 2 mutually exclusive keys above where taking neither in the set is an option, $\times 10$ for the 9 hotbar keys or no hotbar keypress, $\times 2^4$ for the remaining binary 4 keys: `use, drop, attack,` and `jump,` $\times 11^2$ for mouse movements, and finally $+1$ for the `inventory` button which is mutually exclusive with all other actions. $\sim 5.2 \times 10^5$ is still quite large so we chose to implement a hierarchical binary action for camera being moved or not. If this action is on, then there is a secondary discrete action head with 121 classes (the joint distribution of mouse movements because each discretized mouse direction has 11 bins) that determines where to move the mouse. If the hierarchical action is off, then there is no mouse movement, loss for the secondary mouse movement action is masked during training, and the secondary action head need not be sampled during evaluations. While this no longer models the full joint distribution, it is quite a bit better than the factored action space since dependencies between keypresses as well as whether or not to move the mouse (although not which mouse movement) are modeled jointly. The resulting action space has dimension $3^3 \times 10 \times 2^4 \times 2 + 1 = 8461$ (the $11^2$ dimensional multiplier for camera movement has been replaced by a multiplier of 2 here, corresponding to a binary action for whether or not to move the mouse) with an additional 121-dimension head for the joint camera movements. In the future it would be interesting to implement sequential conditional action spaces to more completely model the joint distribution.

In Figure 17 (right) we compare environment rollout performance between BC models with the hierarchical joint action space and with the factored action space. Environment statistics are fairly comparable; however, we see that the factored action space model samples far more null actions. This is an important example of the factored action space failing to correctly model the distribution in the dataset because, due to null action filtering, there are 0 null actions in the dataset these models train on. Despite this, the factored model samples many null actions because the prediction for each key is not conditioned on other keypresses.

### E.4 Foundation Model Training

The foundation model training is similar to the IDM training, with the exception of labels being IDM-generated pseudo labels. The hyperparameters used for foundation model training are listed in Table 4.

| Hyperparameter | Value |
|:---:|:---:|
| Learning rate | 0.002147 |
| Weight decay | 0.0625 |
| Epochs | 30 |
| Batch size | 880 |

Table 4: Hyperparameters for foundation model training

## F   Behavioral Cloning Fine-Tuning

Behavior cloning fine-tuning is similar to the foundation model training, except we either use a focused subset of all the videos (`early_game` dataset, described in A.3) with pseudo labels, or contractor data (`contractor_house` dataset, described in B.4) with ground-truth labels. The hyperparameters used for behavior cloning fine-tuning are listed in Table 5. We used 16 A100 GPUs for about 6 hours when fine-tuning on `contractor_house` dataset, and 16 A100 GPUs for about 2 days when fine-tuning on `early_game` dataset.

| Hyperparameter | Value |
|----------------|----------|
| Learning rate | 0.000181 |
| Weight decay | 0.039428 |
| Epochs | 2 |
| Batch size | 16 |

Table 5: Hyperparameters for behavior cloning fine-tuning

# G  Reinforcement Learning Fine-Tuning

## G.1  Reinforcement Learning Fine-Tuning Training Details

RL experiments were performed with the phasic policy gradient (PPG) algorithm,[65] an RL algorithm based on the proximal policy optimization (PPO) algorithm[81] that increases sample efficiency by performing additional passes over the collected data to optimize the value function as well as an auxiliary value function. These algorithms have been described extensively in previous work,[65,81] so here we describe them only briefly. A major inefficiency when training on-policy algorithms is that, to remain on-policy, one can only take a single gradient step before new rollout data needs to be gathered to continue optimization. To alleviate the potentially destructive effects of taking multiple optimization steps in a single iteration, PPO prevents the policy from changing too much in a single step by clipping the loss when the difference between the current policy and the policy before the update becomes too large.[81] We also use generalized advantage estimation (GAE), which can speed-up credit assignment by looking more than 1 step into the future when determining the advantage of an action, with the look-ahead being determined by hyperparameter $\lambda$.[82]

PPG improves the sample efficiency of PPO when the policy and value function share the same network by following different optimization processes for the policy, the value function, and their shared representation. PPG splits optimization in two phases: a wake phase and a sleep phase. In the wake phase, the policy and value function are optimized as in normal PPO training, with the only exception being that every sample is used at most once, which prevents the policy from overfitting on these samples. In the sleep phase PPG optimizes the value function and an auxiliary value function (which is optimized with the exact same loss as the regular value function, but its output is never used during training), while keeping a Kullback-Leibler (KL) divergence loss to the policy before the start of the sleep phase to ensure that the policy does not change. Because the policy is not optimized in this step, PPG does allow samples to be reused multiple times in this phase. The assumption behind optimizing the value function during the sleep phase is that value function optimization is less sensitive to being trained multiple times on the same sample. Optimizing the auxiliary value function does not directly affect either the value function or the policy, but it can improve the shared representation of both functions (the assumption being that predicting the value-function requires encoding all features that are important for distinguishing states). The coefficients for the three losses (value function loss, auxiliary value function loss, and KL loss) are listed in Table 6. In our experiments a single iteration consists of two sleep cycles and one wake cycle.

Because the value and auxiliary value functions are not optimized during BC pre-training, they are initialized at the start of RL fine-tuning. Each value function is implemented as a single, fully connected layer on top of the last residual transformer block of the pretrained model (Appendix D.1). The weights of the auxiliary value function are randomly initialized while the weights of the regular value function are initialized with zero weights, which appeared to prevent destructive updates early in training that could happen with a randomly initialized value function. To prevent the value-function loss from having gradients that depend greatly on the magnitude of the reward, we normalize the value-function target by subtracting the mean and dividing by the standard deviation, which are estimated through an exponentially weighted moving average.

To prevent catastrophically forgetting the skills of the pretrained network when RL fine-tuning, we apply an auxiliary KL divergence loss between the RL model and the frozen pretrained policy.[11] This loss is defined as:

$$L_{klpt} = \rho \text{KL}(\pi_{pt}, \pi_\theta) \qquad (3)$$

| Hyperparameter | Value |
|---|---|
| Learning rate: | $2 \times 10^{-5}$ |
| Weight decay: | 0.04 |
| Batch size: | 40 |
| Batches per iteration: | 48 |
| Context length: | 128 |
| Discount factor ($\gamma$): | 0.999 |
| GAE $\lambda$: | 0.95 |
| PPO clip: | 0.2 |
| Max Grad norm: | 5 |
| Max Staleness: | 2 |
| PPG sleep cycles: | 2 |
| PPG sleep value-function coefficient: | 0.5 |
| PPG sleep auxiliary value-function coefficient: | 0.5 |
| PPG sleep KL coefficient: | 1.0 |
| PPG sleep max Sample Reuse: | 6 |
| KL divergence coefficient $\rho$: | 0.2 |
| Coefficient $\rho$ decay: | 0.9995 |

Table 6: Hyperparameters for RL experiments. These are the hyperparameters for all treatments with two exceptions. First, when fine-tuning from the early-game model without a KL divergence loss, in addition to the KL divergence loss being set to 0, the learning rate was set to $3 \times 10^{-6}$ (the best setting out of a sweep over 5 different learning rates), as we found that performance was substantially lower with the standard learning rate of $2 \times 10^{-5}$ and the agent did not even learn to collect logs. We suspect that the reason that the learning rate needed to be lowered when fine-tuning without a KL loss is that the KL loss prevents making optimization steps that change the policy too much in a single step, especially in early iterations when the value function has not been optimized yet, and the KL loss thus makes it possible to optimize with a higher learning rate. Second, when running RL from a randomly initialized policy there is no KL divergence loss or KL divergence decay, but instead we use an entropy bonus of 0.01, which reportedly worked well in previous work.[31]

Where $\pi_\theta$ is the the policy being trained, $\pi_{pt}$ is the frozen pretrained policy, $\mathrm{KL}(\pi_{pt}, \pi_\theta)$ is the Kullback-Leibler divergence between the policy being trained and the pretrained policy, and $\rho$ is a coefficient to weight this loss relative to other losses.

In the fine-tuning experiments, this KL divergence loss replaces the common entropy maximization loss, which is often added to RL experiments to encourage exploration.[83,84] The idea behind entropy maximization is that, when all actions appear to have equal value, such as when the agent has not learned about the next reward, it should maximize its entropy to increase the chance that it discovers the next reward. Blindly exploring by maximizing entropy is effective when the state and action spaces are sufficiently small or the reward is sufficiently dense, but becomes infeasible when the state and action spaces are large and rewards are sparse, which is the case in the diamond-pickaxe task. Instead of blindly exploring through uniform-random actions, we assume that the pretrained policy has an action distribution that is much more likely to take sequences of actions that lead to interestingly new states, and thus, in states where the agent assigns equal value to each of its actions, it should mimic the action-distribution of the pretrained policy instead of a uniform-random action distribution. In experiments with a randomly initialized policy we do include the entropy maximization loss with a coefficient of 0.01, which has been an effective setting in other Minecraft work.[31] Empirically, we found that a high coefficient $\rho$ for this KL divergence loss would prevent the agent from properly optimizing the reward function while a low coefficient $\rho$ was ineffective at protecting the learned skills of the pretrained policy and preventing catastrophic forgetting. As such, we start with a relatively high coefficient $\rho$ and decay it by a fixed factor after each iteration (Table 6). This method protects policy skills in early iterations while guaranteeing that the policy can eventually maximize the reward function, regardless of how different its behavior has to be to do so relative to the pretrained policy.

For the reward function we estimated the rough quantities of each item that a human player might gather when trying to craft a diamond pickaxe, and we reward the model for gathering up to that quantity for each item. We started these estimates by iterating over the technology tree backward from

| Item | Quantity rewarded | Reward per item |
|---|---|---|
| Log | 8 | 1/8 |
| Planks | 20 | 1/20 |
| Stick | 16 | 1/16 |
| Crafting table | 1 | 1 |
| Wooden pickaxe | 1 | 1 |
| Cobblestone | 11 | 1/11 |
| Stone pickaxe | 1 | 1 |
| Furnace | 1 | 1 |
| Coal | 5 | 2/5 |
| Torch | 16 | 1/8 |
| Iron ore | 3 | 4/3 |
| Iron ingot | 3 | 4/3 |
| Iron pickaxe | 1 | 4 |
| Diamond | inf | 8/3 |
| Diamond pickaxe | inf | 8 |

Table 7: Reward per item and total quantity rewarded.

a diamond pickaxe and adding the requirements for each item to the reward function (e.g. first we added a diamond pickaxe to the reward function, then we added the 3 diamonds and 2 sticks required for crafting a diamond pickaxe, then we added the 1 iron pickaxe required for mining diamonds, and so on). Then we added coal and torches to the reward function, with coal being useful as fuel when smelting iron and for crafting torches while the torches themselves improve visibility and prevent enemies from spawning. Finally, we reward the model for bringing additional logs (5 logs are required to craft all items in the reward function, but we reward up to 8 logs), which can be used as fuel or crafted into a crafting table or sticks if the agent runs out. In practice the agent rarely collects the additional logs, places the torches, or uses coal as fuel when smelting, but the reward function was based on human expectations on what would be useful to execute this task, rather than designed around how an RL model behaves after training. Finally, to encourage the agent to keep mining diamonds and crafting diamond pickaxes after it has crafted its first diamond pickaxe, we did not put a limit on the number of diamonds or diamond pickaxes that would be rewarded.

The rewards for the different items are separated into 4 tiers, roughly depending on how late a player would usually get the relevant item. The first tier consists of all wooden and stone items and has a base reward of 1, the second tier consists of all items requiring coal with a base reward of 2, the third tier consists of all items requiring iron with a base reward of 4, and the final tier is diamond with a base reward of 8. Thus items later in the sequence of items towards a diamond pickaxe generally give a higher reward. To make sure that the agent does not over-value items that are supposed to be gathered in bulk (e.g. the agent is rewarded for up to 20 planks but only up to 1 crafting table, which can cause the agent to focus on planks at the expense of creating a crafting table), we divide the base reward of each item by the total quantity that the agent gets rewarded for (for the purpose of determining the reward, the total quantity for diamonds is 3 and the total quantity for diamond pickaxes is 1, even though we did not put a limit on the number of these items being rewarded). For example, the agent is rewarded for 3 iron ore, which has a base reward of 4 for being in the iron tier and up to 3 blocks of iron ore are rewarded, thus the reward per block of iron ore is $4/3$. The quantity and reward for each item are listed in Table 7.

While every item in the sequence towards a diamond pickaxe is rewarded, the reward function is still sparse and, in some cases, even deceptive. The sparsity comes from the fact that it can take thousands of actions to find the next reward, even after the agent has acquired all the necessary prerequisites (e.g. human players often take more than 10,000 actions to find a diamond after crafting an iron pickaxe). The reward function can be deceptive when the most efficient method for getting one item can make it far more difficult to get the next item. For example, a good strategy for the agent to craft a stone pickaxe quickly is to mine (i.e. spend a few seconds to pick up) its crafting table after crafting a wooden pickaxe, such that the agent has immediate access to a crafting table as soon as it has collected enough cobblestone. However, the fastest way to get a reward for gathering cobblestone is to mine down immediately after crafting a wooden pickaxe, while leaving the crafting table behind.

Thus following the optimal strategy for gathering cobblestone makes it more difficult to learn to craft a stone pickaxe.

Experiments ran for approximately 6 days (144 hours) on 80 GPUs (for policy optimization) and 56,719 CPUs (mostly for collecting rollouts from Minecraft). In this time the algorithm performed roughly 4,000 optimization iterations and collected roughly 1.4 million Minecraft episodes consisting of 12,000 frames each, for a total of 16.8 billion frames.

## G.2 Reinforcement Learning Fine-Tuning Additional Data

Additional figures that are helpful for understanding the main results of the RL fine-tuning experiments are presented in this section. First, we show the items-over-training figure when RL fine-tuning from the early-game model without a KL loss (Fig. 18). When training without a KL loss, the model only learns to obtain the four items that the early-game model is capable of getting zero-shot, which are logs, planks, sticks, and crafting tables. Second, we present preliminary experiments in which we directly compare RL fine-tuning from the house-building model and RL fine-tuning from the early-game model (Fig. 19). These experiments differ from the main experiments in that, for both treatments shown here, the KL loss coefficient was set to $0.4$, the learning rate was set to $6 \times 10^{-5}$, and the reward for each item was $1/quantity$ for all items (i.e. items closer to the diamond pickaxe did not have an increased reward). While RL fine-tuning from the house-building model initially worked better than RL fine-tuning from the early-game model, fine-tuning from the early-game model worked better after 800,000 episodes and showed signs of smelting iron ingots, which is why the early-game model was chosen for the main experiments.

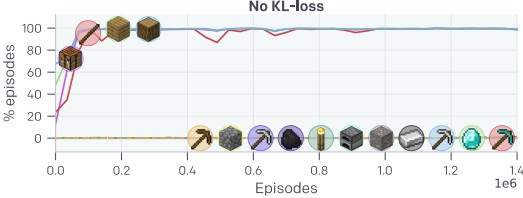

Figure 18: Items obtained when RL fine-tuning from the early-game model without a KL loss. The model learns to obtain all items that the early-game model can craft zero-shot, which are logs, planks, sticks, and a crafting table. In contrast to the treatment with a KL-penalty, it does not learn any items beyond these initial four, likely because skills that are not performed zero-shot, and for which the model thus does not initially see any reward, are catastrophically forgotten while the first four items are learned.

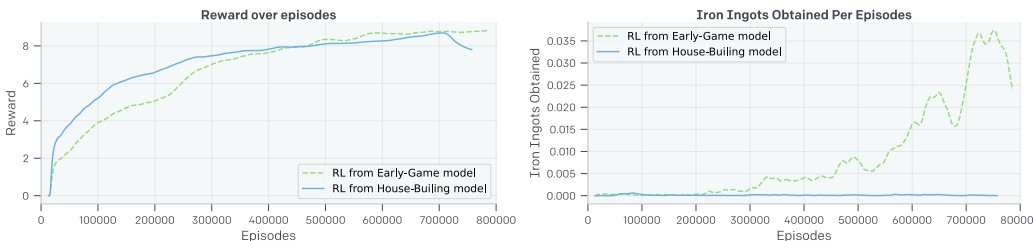

Figure 19: Preliminary experiments when RL fine-tuning from the early-game model compared to RL fine-tuning from the house-building model. **(Left)** While reward initially increases faster when fine-tuning from the house-building model, fine-tuning form the early-game model eventually obtains a slightly higher reward. **(Right)** RL fine-tuning from the early-game model has a higher likelihood of smelting an iron-ingot, which is why the early-game model was chosen for future RL fine-tuning experiments.

# H Foundation Model Scaling

In early experiments we found that increasing model size led to models staying in the efficient learning regime longer into training.[64] Here we compare the 0.5B model described in Section 4.2 to both a 248M and 71M parameter model. Both of these models are trained for 15 epochs as compared to the 30 epochs the 0.5B model trained for. These models have the same architecture as the 0.5B model but each layer in the 248M parameter model has 1/2 the width and each layer in the 71M parameter model 1/3 the width. The 71M model was trained with an initial learning rate of 0.001586, batch size of 480, and weight decay of 0.044506. The 248M model had an initial learning rate of 0.001831, batch size of 640, and weight decay of 0.051376.

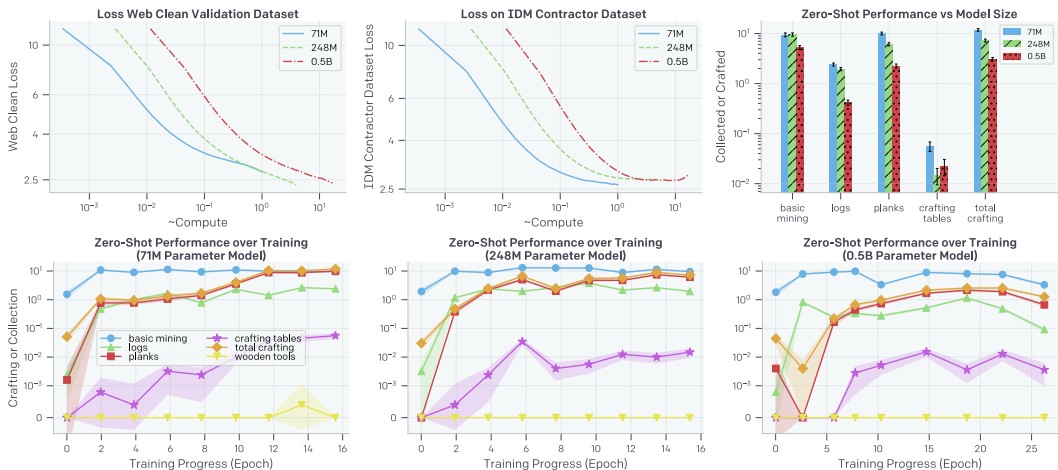

Figure 20: Training and Zero-Shot Performance versus Model Scale. In the first two plots the x-axis is compute normalized to that used by the 71M parameter model, such that after 15 epochs of training the 71M model has used 1 "compute". The 248M parameter model and the 71M model are trained on the same amount of data (15 epochs), and the 0.5B parameter model is trained on 30 epochs of data. **(Top Left)** Loss on the `web_clean` validation dataset. **(Top Middle)** Loss on the IDM contractor dataset; note that these models were trained only on `web_clean` and not on any contractor data. **(Top Right)** Zero-shot environment rollout performance at the end of training. **(Bottom)** Zero-shot environment rollout performance over training for the 71M model (bottom left), 248M model (bottom middle), and 0.5B model (bottom right).

In Figure 20 we show validation loss on `web_clean` with IDM pseudo-labels, loss on the contractor dataset used to train the IDM with ground truth labels collected during contractor play, and zero-shot environment performance for the 71M, 248M, and 0.5B models. While larger models have better validation loss on `web_clean`, these results do not tell the clear story that the 0.5B model is better than its smaller counterparts. The 71M model has the lowest contractor dataset loss while having the highest `web_clean` loss, and it also has the best zero-shot environment performance. In fact, we see that the 71M model even had non-zero wooden tool crafting (Fig. 20 bottom left). The 248M model also appears to be better at crafting than the 0.5B, and also has lower contractor dataset loss.

While the zero-shot results suggest smaller models are better, fine-tuning tells another story. When fine-tuning to `contractor_house`, model size rank ordering reverses and now the 0.5B model performs best both in validation loss (Fig. 21 left) and in environment performance (Fig. 21 right) followed by the 248M model and then the 71M model. Environment model rollouts are performed using the same game engine that we use to collect contractor data, which could be visually distinct from videos taken from the web. It is plausible that the larger models overfocus on the visual peculiarities in web data during pretraining since they have worse contractor data loss (Fig.20 top middle), and this causes them to perform more poorly in the environment zero-shot. However, we hypothesize that because the `contractor_house` dataset we fine-tune to is collected from our game engine, the larger models that are a better overall Minecraft prior (as indicated by lower `web_clean` validation loss in Fig.20 top left) can quickly shift their low level features to perform better on data coming from our game engine, resulting in better environment rollout performance. This hypothesis is further supported by Fig. 21 (middle) showing loss on the contractor dataset collected for IDM

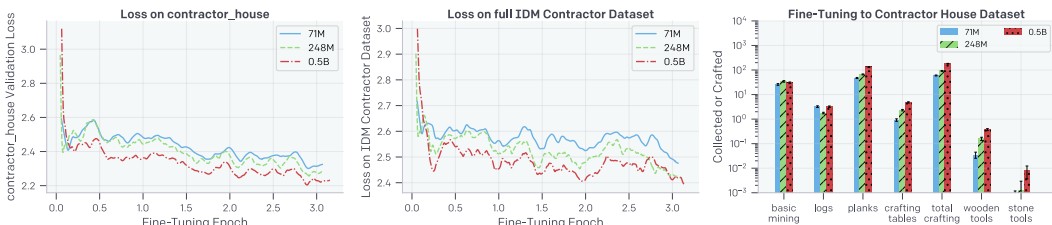

Figure 21: Fine-tuning VPT foundation models with increasing model sizes to the `contractor_house` dataset. **(Left)** Loss on the `contractor_house` holdout validation set. **(Middle)** Loss on the full contractor dataset collected to train the IDM; this dataset is disjoint from `contractor_house`. **(Right)** Environment rollout performance at the end of fine-tuning.

training, which has no overlap with `contractor_house`. After just a few steps of fine-tuning to `contractor_house`, all models quickly improve in loss on the full IDM contractor dataset, with larger models now performing best. While not conclusive, we believe this investigation provides some intuition for future studies of model scaling for sequential decision making problems.

## I   Text Conditioning

Goal-conditioned policies[85,86] make it possible for a single agent to perform a wide variety of goals in a single environment, which is particularly relevant in open-ended environments such as Minecraft. In recent work, goal specification has increasingly taken the form of domain specific languages[87], or even natural language[88,89]. The benefits of language-conditioned agents can be tremendous, especially *natural*-language-conditioned agents, as their goal space contains a wide variety of potentially very complex tasks. Text conditional models have shown an amazing ability to perform tasks zero-shot (or learn them few-shot) including generalizing in impressive ways via the compositional and combinatorial possibilities allowed by natural language (e.g. GPT[1] and DALL·E 2[90]). We hypothesize that we should expect similar capabilities to emerge with natural-language-conditioned virtual agents, if they are similarly trained on enormous amounts of data (that goes from a natural language description to a sequence of actions that completes the specified goal). In this section we take preliminary steps toward that future. Our preliminary experiments provide evidence that it is possible to pretrain a natural-language-conditioned model for Minecraft using the general approach presented in this paper (VPT) plus conditioning on the speech that often accompanies videos.

In online videos, the human actor sometimes indicates their intent in their verbal commentary (e.g. "Let's go chop some trees to make a wooden axe" or "now let's learn how to crop photos in Photoshop"). Conditioning on this closed caption data could produce a *steerable* pre-trained model: i.e., it may later be possible to condition the model with text such as "I am going to craft a wooden pickaxe" or "I am going to build a house," and have the agent perform those tasks specifically rather than simply follow typical human behavior (as was investigated in the rest of this paper). An alternate way to produce a steerable agent is via RL fine-tuning, which we could have done in Section 4.4 by adding a bit indicating the task to be completed, as has been done in prior work[31]. However, conditioning on natural language offers many benefits over that approach. First, it is flexible and powerful, being able to express any task. Second, one does not need to preconceive of the task to be completed ahead of time. This would allow for general, capable, zero-shot agents like GPT, but extending those capabilities to embodied tasks such as completing tasks on computers or in simulated 3D worlds. Third, text conditioning can be used even when tasks are difficult to specify via reward functions (e.g. "Let's build a house" or–if the agent is capable of doing it–more complex things like "I will now build a castle surrounded by a moat"). In the limit, VPT+text could conceivably produce powerful, capable, natural-language-conditional agents with the powers of GPT to meta-learn, follow instructions, and complete tasks zero or few shot, but in the form of agents that can act in virtual worlds, complete tasks on computers, and in other similar embodied sequential decision domains. We do not reach those lofty goals in this work, but we began a first step towards exploring in that direction.

Many Minecraft videos feature audio commentary from the player. This commentary is sometimes present in the form of closed captions for the videos, or could be extracted post-hoc using automated speech recognition (ASR).[91] Our dataset features about 17k hours of content with associated closed captions.

We fine-tuned the 220 million parameter VPT foundation model used in the RL-fine-tuning experiments (chosen vs. 0.5B for the same reason: to reduce compute costs) with an additional text-conditioning input on the subset of our data for which closed captions are available. To obtain the conditioning input, we first split videos into 30 second chunks. The same text is associated with every frame in a given chunk, and is made up of all the closed captions occurring within that chunk, as well as the line of text preceding and following the chunk (if any). Because the vast majority (around 95%) of our closed caption data lacks capitalization and punctuation, it is punctuated using the rpunct library[92]. We then obtain a text embedding vector of length 4,096 from the OpenAI embedding API[93], which is processed by a randomly initialized multi-layer perceptron (MLP) with two hidden layers of size 2,048. The resulting activations are added for each frame to the pretrained model activations before the transformer layers (`pretransformerActivations += mlp(textEmbedding)`). The model is fine-tuned for four epochs.

| Variant name | String |
|:---:|:---:|
| dig | I'm going to dig as far as possible |
| dirt | I'm going to collect dirt |
| explore | I'm going to explore |
| house | I'm going to make a house |
| seed | I'm going to collect seeds |
| water | I'm going to find water |
| wood | I'm going to chop wood |

Table 8: Strings corresponding to each conditioning variant.

Our model shows evidence of steerability. When conditioned on sentences that incite the agent to explore (such as "I'm going to explore" and "I'm going to find water") the agent travels significantly farther from its spawn point (Figure 22a). Additionally, we can steer the agent to preferentially collect early game items such as seeds, wood, and dirt by conditioning with text such as "I'm going to collect seeds/chop wood/collect dirt" (Figure 22b,c,d).

While our results show some level of steerability, more work is required to increase it. For example, we were not able to successfully steer agents to gather flowers or to hunt, both of which are possible in the early game, but less common (and, in the case of hunting animals, much more difficult) than gathering dirt, wood, or seeds. Likewise, an experiment in which the agent is presented with a crafting window and various resources, and conditioned to craft a given item (e.g. "I'm going to craft a wooden axe") failed to show that the conditioning had a significant effect on which items got crafted. Instead, it seemed the agent was more influenced by the prior, unconditional probability of what human players would craft next given the resources available, which is not too surprising since in Minecraft, especially in the early game, there is a relatively consistent path to gathering resources in a specific order go produce more powerful tools (Fig. 6). For example, if the agent had the resources to make a stone pickaxe and we asked it instead to make a (weaker) wooden pickaxe, it often would make the stone pickaxe anyway. Finally, looking at videos of agent behaviors failed to convince us that the "house" conditioning causes the agents to take more steps towards building a house than other variants.

Thus, our results show that it is possible to train a somewhat steerable natural-language-conditioned agent. However, its steerability is still too weak to be practically useful, and it is far from what we believe could be accomplished with more research, data, and training compute. Another exciting research direction is to have the model predict future text as well as just the next action.

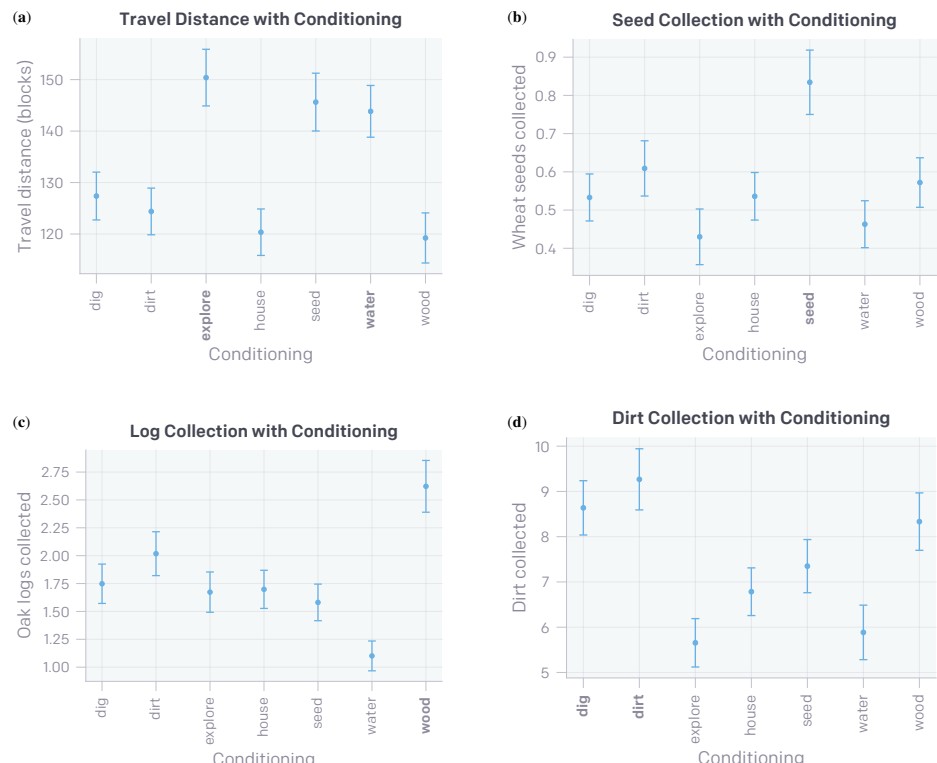

Figure 22: Evidence for conditioning. In each plot, the variants expected to stand out are shown in bold. The strings corresponding to each variant are shown in Table 8. Statistics are measured over 5 minute episodes. **(a)** Distance traveled by the agent . Both "explore" and "water" text strings should encourage a steerable agent to move more than when doing other tasks, which is what occurs. Grass (which is needed to get seeds) is not present in all biomes, which is likely why the "seed" condition produces more travel (as the agent sometimes needs to move to a biome with grass). The travel distance is the Euclidean distance from the spawn point to the farthest point the agent reached during the episode on the horizontal (x-z) plane. **(b)** Collection of wheat seeds. The "seed" variant collects substantially more than other variants, as expected of a steerable agent. **(c)** Collection of oak (the most common type of wood) logs. The "wood" variant collects significantly more oak logs, as is to be expected of a steerable agent (we speculate that the "water" variant collects less because there are no trees in water). **(d)** Collection of dirt. The "dirt" and "dig" variants collect a large amount, and are the variants that are (indirectly in the case of "dig") conditioned to collect dirt. It is easy to mistakenly aim at the ground rather than at grass or trees when collecting seeds or wood, which likely explains the slightly higher amount of dirt collected by these variants. In all cases, the error bars are 95% confidence intervals of the mean, over 1,000 episodes per conditioning variant. Treatments for which the bars in each bar plot do not overlap are statistically significantly different at a $p < 0.05$ level.