# OpenReview forum: "Video PreTraining (VPT): Learning to Act by Watching Unlabeled Online Videos"
_NeurIPS.cc/2022/Conference — NeurIPS 2022 Accept_

### Official Review · Reviewer_Q4Hg · 2022-06-29

**Rating:** 6
**Confidence:** 5
**Soundness:** 2 fair
**Presentation:** 3 good
**Contribution:** 2 fair

**Summary:**

In this paper, the authors use a small amount of labeled data and a large amount of unlabelled data for solving Minecraft tasks. The authors aim to build a general-purpose framework for sequential decision-making tasks. The model has zero-shot generalization ability.

**Questions:**

Using large models and massive data always seems helpful for better results and zero-shot generalization. This has become an obvious "fact" based on recent papers, such as DALLE [4], CLIP[3], LID[1], and GATO[2]. So what are the main contribution or novelty left after excluding the large models and massive data part?

This is a good paper, trying to solve a challenging problem. Minecraft tasks are challenging, and there are no methods that can solve such challenging tasks. However, if the good performance mostly comes from the usage of large data and large models, this paper is not convincing enough to be accepted. If the authors could provide more evidence of what makes this paper different from other decision-making works in terms of the method part (except for the large data part), I would change my mind.

[3] Learning Transferable Visual Models From Natural Language Supervision
[4] Hierarchical Text-Conditional Image Generation with CLIP Latents


**Limitations:**

(-) The results highly depend on the policy trained on the small amount of labeled data. If this policy is not good enough, it cannot provide reliable predictions for the unlabelled data.

(-) The unlabelled video data cannot be used directly. The authors still need carefully design the filtering procedure to ensure the data are clean enough for training a better policy.

(-) This is a good paper with impressive results, but it seems the novelty is limited and not a good match for NeurIPS.

**Strengths And Weaknesses:**

(+) The results are impressive. Minecraft tasks are challenging. This paper shows a promising way to solve such tasks.

(-) The idea is not novel. The action is predicted based on a set of historical observations in Eqn 1. Even though the method part has three stages as shown in Fig2, the technical contribution is limited in the method part.

(-) This paper did not cite works that are quite strongly related to this paper, such as [1], [2], and [3].
[1] Pre-Trained Language Models for Interactive Decision-Making
[2] A Generalist Agent
[3] Can Wikipedia Help Offline Reinforcement Learning?

---

> ### Author Response · Authors · 2022-08-02
> **Response**
>
> Thank you for your review and for noting that our results are impressive and recognize that no prior work has been able to match our results in the difficult domain of Minecraft. We’ll first respond to your largest criticism around novelty and technical contribution and then address the remaining more minor points. Your overall review of our work is quite different from that of the other reviewers, so we are interested to better understand and address the limitations you feel the work has. We are glad to hear that you are open to changing your mind about your score for the paper and engaging with us and the other reviewers in the review process.
>
> The point of this paper is not that more data = better performance, which as you point out has been validated many times over in multiple domains. The point of this paper is that for domains like Minecraft or e.g. computer usage, a huge amount of data exists but it does not have action labels, so it is impossible to train a behavioral prior using it without additional scientific tools. Our method aims to unlock this data for use in constructing behavioral priors with a simple method: labeling it with an inverse dynamics model. To the best of our knowledge, we believe that this method is novel, but if you could point us to a paper that uses the same method we would love to read it and cite it.
>
> “This paper did not cite works that are quite strongly related to this paper”
>  - LID and Can Wikipedia Help Offline Reinforcement Learning? - In our opinion these papers are only somewhat related in that they use a pretrained model to solve reinforcement learning tasks. They do so by hypothesizing that language models are good priors for non-language sequence tasks and then embed observations from the RL environment in a way the language model can accept. While this does not solve the issue that VPT solves (which again is to unlock unlabeled data for use in pretraining behavioral priors), they could potentially complement our method by initializing the VPT foundation model or IDM with a reasonable representational prior. However, even if we did this we would still have to label the unlabeled internet data in order to train on it and create a behavioral prior, which is why we feel these papers are not strongly related. Thus we feel our paper is distinct and an essential independent contribution. We’ll add a citation for these works and describe how they are related and can be combined in the Discussion section of our work. Thank you for the suggestion!
> GATO. This paper was only released 7 days prior to the Neurips submission, and was not peer reviewed at the time. More importantly, we believe it is not directly related to the main contribution of our work. For all control tasks, GATO only trains on labeled data. The point of our work is not to prove that more labeled data is better, but rather can we train an effective behavioral prior from a large amount of unlabeled data. We will add a citation to this paper in the introduction.
>
> “The results highly depend on the policy trained on the small amount of labeled data. If this policy is not good enough, it cannot provide reliable predictions for the unlabelled data.”
>  - We agree this is a central hypothesis of our paper (that a small amount of data is sufficient for training an IDM accurate enough to label a large unlabeled dataset sufficiently well such that BC-training on that labeled data produces good behaviors). We feel the zero-shot and fine-tuning results, including those with reinforcement learning that produced unprecedented new capabilities for computer agents in Minecraft, confirm that this hypothesis is true. Moreover, we provide experiments showing just how little labeled contractor data may be required in our domain (Section 4.6).
>
> “The unlabelled video data cannot be used directly. The authors still need carefully design the filtering procedure to ensure the data are clean enough for training a better policy.”
>  - Good question! We’ve ablated the data cleaning process at request of another reviewer. That being said, we tried to limit ourselves to simple data cleaning steps and actually we did not iterate on the data cleaning process over the course of the project; it was the first process we tried. We will make this more clear in the paper text. Also please see Appendix A.2.3 of the updated text if you are interested in the effect of data cleaning. We found that it is indeed beneficial to clean the data in this way; however, again this was a fairly simple data cleaning process that we did not tune.
>
> Thank you for your thoughtful commentary. We hope we have addressed your concerns and–as you said– you are open to changing your score!

---

> > ### Comment · Reviewer_Q4Hg · 2022-08-04
> > **Response to authors**
> >
> > The response addressed some of my concerns. It would be good if they included the discussion of the mentioned related works in their final version.

---

> > > ### Author Response · Authors · 2022-08-05
> > > **Response**
> > >
> > > Thank you for increasing your score! You mention that we only address some of your concerns; which concerns do you still have? We would be happy to discuss them further with an eye towards understanding where the remaining gap lies between your score and the scores of other reviewers.
> > >
> > > In our updated draft we posted Tuesday, at your suggestion we cite _GATO_ in the Introduction, and we cite _LID_ and _Can Wikipedia Help Offline Reinforcement Learning?_ in the Discussion section. We already had cited _CLIP_ in the original submission, and we felt that _DALL-E_ was not relevant enough to cite in this work because arguably we already cite stronger cases showing that “more labeled data is better” such as _GPT-3_. We would love to hear if you think we could improve our discussion of these works or think otherwise in the case of _DALL-E_.

---

> > > > ### Comment · Reviewer_Q4Hg · 2022-08-05
> > > > **Response**
> > > >
> > > > I agree that this paper solves a challenging task. But I am not convinced that the inverse dynamics model is a novel technical contribution based on your reply `Our method aims to unlock this data for use in constructing behavioral priors with a simple method: labeling it with an inverse dynamics model. To the best of our knowledge, we believe that this method is novel.` Why do you think the inverse dynamics model is novel?
> > > >
> > > > Does the `small amount of labelled data` already see all the actions, observations, and objects? What if the actions, observations, and objects are out of the training distribution (not seen during training)?
> > > >
> > > > I feel this paper is solving a challenging problem and I give an Accept, but it also requires lots of manually designed data collection and data filtering, and the inverse dynamics model and policy learning part seem not very novel.

---

> > > > > ### Author Response · Authors · 2022-08-08
> > > > > **Response**
> > > > >
> > > > > Thank you for responding quickly! We do not think that our inverse dynamics models are novel, and we did not claim as such in the paper or in our responses to your questions. We define our method to be the entirety of Figure 2, which includes collecting contractor data to train an IDM, using the IDM to label a large amount of unlabeled data, and then finally training a policy on it with behavioral cloning. While each independent part of the method is not novel, we think their combination is. The NeurIPS reviewer guidelines around originality are: _“Are the tasks or methods new? Is the work a novel combination of well-known techniques? (This can be valuable!)”_, and in our case we feel that we show that this novel combination is valuable and effective with unprecedented results in Minecraft.
> > > > >
> > > > > “Does the small amount of labelled data already see all the actions, observations, and objects? What if the actions, observations, and objects are out of the training distribution?”
> > > > >  - We shared your concern at the start of the project; however, in general human contractors tend to reach most parts of the game state space, and so we did not find this to be an issue. This shouldn’t be surprising since the distribution of unlabeled data we collect from the internet is also human play. In our original submission we wrote, “Because human contractors reach most relevant parts of the state space, we can hold the IDM fixed throughout BC training.”
> > > > >
> > > > > “I feel this paper is solving a challenging problem and I give an Accept, but it also requires lots of manually designed data collection and data filtering, and the inverse dynamics model and policy learning part seem not very novel.”
> > > > >  - Manually designed data collection and data filtering. We did not do anything out of the ordinary or arduous here, and we do not think that collecting data should count against a method, especially when the data collection was not complicated. Certainly we would also prefer it if one were able to obtain results as good as ours without any data collection; however, as far as we are aware it is nigh impossible to do so with RL or any other method. Finally, we are open sourcing all of the data we did collect for use in future projects.
> > > > >  - Again we do not claim that our inverse dynamics model or policy learning are novel in isolation, but we do think that their combination with easy-to-collect contractor data is novel and effective.
> > > > >
> > > > > We are also curious about your ratings for the soundness (2 out of 4). Could you elaborate why you think our experiments are unsound?

---

### Official Review · Reviewer_uuCc · 2022-07-09

**Rating:** 9
**Confidence:** 4
**Soundness:** 4 excellent
**Presentation:** 4 excellent
**Contribution:** 4 excellent

**Summary:**

In this paper, the authors use ~2,000 hours of hand-labeled data to train an Inverse Dynamics Model (IDM) to predict the mouse and keyboard actions from raw video of Minecraft games. This model benefits from using both the future and past video frames when computing per-frame predictions (i.e. “acausality”).

The authors then use this "IDM" to predict the mouse and keyboard action labels for 70,000 hours of raw unlabelled video of Minecraft games sampled from the internet. The authors call the labels predicted by the IDM on this data “pseudo-labels”.

Given the 70,000 hours of pseudo-labelled gameplay, the authors train a model to predict future actions from past actions using Behavioural Cloning (BC). This model is referred to throughout the work as a Video PreTraining, or VPT model.

The authors show that their VPT model already achieves “non-trivial zero shot performance”, and has thus successfully learnt a prior over the behaviour distribution. To achieve their strongest results, the authors then incorporated this VPT into two further downstream experiments: fine tuning using more specialised datasets covering a specific desired behaviour, and applying reinforcement learning.

The authors found that factoring the problem into two parts: an inverse dynamics model to predict pseudo labels, and a model for behavioural cloning, were key to achieving good performance. The VPT model trained by the authors using this technique is able to provide a very strong baseline behavioural prior, and further fine tuning allows the authors to obtain very compelling results in Minecraft gameplay that appear far beyond the current capabilities of RL to learn from scratch. Perhaps the most impressive demonstration of results is that the agent was able to create "diamond tools", which require tens of thousands of complex actions be taken by the agent.


**Questions:**

* Deep in the appendix it is mentioned that the BC model and the IDM are identical. This information would be useful in the main body of the paper. i.e. - “The behavioural cloning model architecture is the same as the IDM architecture described in Appendix 971 D.1 except that we modify the architecture so that it is causal”. Minor nit: in the appendix, it is written “residual transformer layers”, where I believe these are  referred to as attention layers in the main body. (attention layers seemed clearer.

* Minor nit: I didn’t really understand figure 3 (right) - what is being compared exactly here?

* Minor nit: IDM Architecture [appendix] - there’s a long wordy description of the model, but maybe a picture or table could be nice?

* Minor nit: Figure 13 (and others) in the appendix appear almost “washed out” :)

**Limitations:**

The authors could certainly elaborate further on the generalisability of the methods presented - how important is it that e.g. screen resolution and settings match up so closely? To what extent could we hope to learn an inverse kinematics model on a more general or noisy distribution of data?

For example, it seems unlikely to me that this could currently work on real-world video (which would be an exciting prospect for e.g. autonomous-driving). Generally, the "results and conclusions" paragraph could benefit from more thought and further deeper discussion.

**Strengths And Weaknesses:**

Strengths:
* The work is very well-executed and demonstrates very impressive results on a challenging learning task, namely Minecraft with the full action space. The excellent performance through training a large behavioural prior is the paper's main novelty, and this largely results from the author's careful execution, and careful application of learning methods. This also clearly required resolving numerous technical challenges, which are mostly enumerated in depth in the supplemental material.
* The outlined method could be generalised to other learning tasks, unlocking the potential of large amounts of unlabelled video data that is increasingly available. This would not be without challenges (inverse dynamics is hard), but the approach seems promising and worthy of further study.
* The paper is clearly written, making it easy to follow and understand the work. The level of detail is appropriate across the sections, and the supplemental material appears to contain sufficient details required to reproduce the work.

Weaknesses:
* The described method may currently be limited in its generality - prediction of inverse dynamics is a difficult problem in its own right, and is somewhat simplified by the world studied by the authors - though this is a compelling direction for further research.
* It would be nice to see a little more depth in the results and conclusions section, and more discussion of future work.
* A little more detail about the architectures and models in the main body of the paper could be welcome.

---

> ### Author Response · Authors · 2022-08-02
> **Response**
>
> Thank you for your thoughtful review. We’re glad that you felt our work was well executed, well presented, and has potential for generalizability to other domains! We will respond to your criticisms and questions in turn.
>
> “The described method may currently be limited in its generality - prediction of inverse dynamics is a difficult problem in its own right, and is somewhat simplified by the world studied by the authors - though this is a compelling direction for further research.” and “The authors could certainly elaborate further on the generalisability of the methods presented - how important is it that e.g. screen resolution and settings match up so closely? To what extent could we hope to learn an inverse kinematics model on a more general or noisy distribution of data?”
>  - We only experiment in a domain where there is a wealth of first person (and therefore aligned to how the agent would act) demonstrations.  As we note, there are many other domains where it would be possible to get this type of data (e.g. computer usage). The videos of Minecraft in our dataset that were uploaded to the internet do not have consistent screen resolutions and graphic settings, so our work already provides some evidence that VPT can work in noisy settings. We  agree it is an extremely interesting line of future research to try VPT or similar methods in even noisier, more general, or even unaligned domains. We will add more discussion of this into the Discussion section. Thank you for the suggestion!
>
> “It would be nice to see a little more depth in the results and conclusions section, and more discussion of future work.”
> - We will add more, and have received some great suggestions from you and the other reviewers!
>
> “A little more detail about the architectures and models in the main body of the paper could be welcome.”
>  - These were not novel nor the driving force behind the success of the method, and unfortunately we’re already at the limit of the NeurIPS space constraints so we won’t be able to move these details to the main paper body.
>
> Nits - thank you! We’ll fix as many of these as we can in the allotted time.

---

### Official Review · Reviewer_m9Gz · 2022-07-10

**Rating:** 9
**Confidence:** 4
**Soundness:** 4 excellent
**Presentation:** 4 excellent
**Contribution:** 4 excellent

**Summary:**

This paper introduces Video Pre-training (VPT), a foundation model pre-trained on large-scale unlabelled videos in Minecraft along with a small amount of human demonstrations. The small amount of data is used to train an inverse dynamics model that can identify the action (M&K inputs directly which is quite neat) taken between frames, and can be used to relabel the larger unlabelled subset. VPT shows some promising zero-shot behaviour, but its performance is significantly improved by fine-tuning with Reinforcement Learning.

The experiments in this paper are of an unprecedented scale in RL, performed on a popular/relevant domain of Minecraft where the range of possible capabilities are numerous, and has large implications for the RL field in general. More generally, this paper points towards a unifying scale and fine-tune paradigm for RL. A large scale foundation model learns a "common sense" layer for the agents, and the agents can be later on made goal-directed by fine-tuning on a relevant reward function. The bitter lesson strikes again as we see that to get to agents with such general capabilities, all we needed was a clean large-scale dataset and simple/existing methods like IDM and PPO sufficed.

**Questions:**

* The decreasing zero-shot performance with increasing model scale seems puzzling. Why do you think that happens to be the case? Are the model under-trained?
* How were the optimal model/data/compute scales determined? Were these ad-hoc choices or did you refer to Kaplan or Chinchilla scaling laws?



**Limitations:**

The limitations should point out that a VPT like recipe would only work on RL domains where large-scale unlabelled data is accessible. It should also point to the reliance of carefully designed reward functions.

**Strengths And Weaknesses:**

Strengths:
* The experiments in this paper are of an unprecedented scale in RL, performed on a popular/relevant domain of Minecraft where the range of possible capabilities are numerous, and has large implications for the way RL field in general. More generally, this paper points towards a  unifying scale and fine-tune paradigm for RL.
* It uses the native mouse and keyboard interface for its agents, thus the results in this paper should translate well to other domains as well.
* The trained agents are able to demonstrate very sophisticated capabilities such as crafting diamonds which seemed quite out of reach for AI agents until now.

Weaknesses:
* Because of the scale of experiments, the design choices are not carefully ablated. Is IDM the way to learn from contractor videos, we discard a ton of info. Similarly, is predicting only actions in VPT optimal (how about tokenized frame predictions)? PPO seems to be a legacy choice as well, and we don't get to see how it performs other off-policy or model-based methods.
* Reward functions are very carefully human designed to get the right behaviour. This seems antithetical to the scaling paradigm, at internet scale, it should be possible to infer reward functions directly.

---

> ### Author Response · Authors · 2022-08-02
> **Response**
>
> Thank you for your thoughtful review. We’re glad that you agree that this framework could be paradigm-shifting for the RL community! We’ll respond to each of the weaknesses and questions you raise in turn.
>
> “Is IDM the way to learn from contractor videos, we discard a ton of info. Similarly, is predicting only actions in VPT optimal”
>  - Great question. Behavioral prior’s must predict actions by definition, otherwise they wouldn’t be able to act in the environment (this is in contrast to what we call "representational priors", such as might be achieved by predicting future frames). In this work we wanted to focus on constructing behavioral priors during large-scale pretraining, and we found that action prediction was sufficient to obtain our results. Due to the large design space of auxiliary objectives, e.g. next frame prediction as you suggest, image auto-encoding, video auto-encoding, time contrastive losses, CLIP style losses versus closed captions, etc, we felt that investigating all of these were outside the scope of this already rather beefy paper. That being said, we agree it is a fascinating direction for future work, and we will add some discussion around this in the Discussion section.
>
> “PPO seems to be a legacy choice as well”
>  - We actually use the PPG (Phasic Policy Gradient) algorithm, which is based on PPO. We decided to use an "off the shelf" RL algorithm like PPG because the specific choice of RL algorithm was not the focus of this paper. RL is just one way to fine-tune such a behavioral prior (we also discuss behavioral cloning fine-tuning in the paper). Using a common and well understood RL algorithm makes it more clear that our results are due to the VPT prior and not due to a more complex RL algorithm. For these reasons we felt that investigating off-policy, model-based, or other RL algorithms was outside the scope of this paper. That being said, we absolutely agree that this is an interesting direction for future research, and we will note as such in the Discussion section as an potentially fruitful avenue for future research.
>
> “Reward functions are very carefully human designed to get the right behaviour”
>  - The purpose of our work was not to introduce a general reward function, but rather to introduce a method to generate a prior that can be then used to help solve downstream tasks. Our reward function, while hand-engineered, is still extremely sparse making it a good benchmark for VPT, and as we show in the paper it is such a difficult reward function to optimize that RL from scratch can make almost no progress on it. Furthermore, we actually spent very little time tuning the reward function as it is quite a natural reward function in Minecraft (rewarding items in the technology tree path on the way to the target item). That being said, we absolutely agree that works attempting to construct more general reward functions are extremely promising and we’ll note this in the Discussion section, e.g. there was work released after the neurips deadline called MineDojo that does exactly this and could be complementary to VPT.
>
> “The decreasing zero-shot performance with increasing model scale seems puzzling. Why do you think that happens to be the case? Are the model under-trained?”
>  - Great question! We were also puzzled by this and already provide some discussion around this in Appendix Section H. We see there that larger models (and therefore models with lower loss) have worse zero-shot performance; however, they do perform better when fine-tuned to our contractor_house dataset. While that analysis is not definitive, it points to the possibility that our models start to overfit to some visual peculiarities in the internet dataset that are not present in our environment, but then when presented with data from our environment they can quickly update this high frequency features. We’ll make sure we point more directly to this discussion from the main body of the paper..
>
> “How were the optimal model/data/compute scales determined? Were these ad-hoc choices or did you refer to Kaplan or Chinchilla scaling laws?”
>  - We were inspired by these works but due to compute limitations we were unable to run comprehensive studies to fit scaling laws for this domain. For model size, we noticed that the 248M parameter model at some point during training started to go into a less compute efficient regime (shallower slope on a loss versus compute log-log plot), so we decided to train the 0.5B parameter model. We tried to tune model size to the dataset size we had such that we could use all of it, and we show a variety of datascale ablations throughout the paper. We will update the text to make this more clear.
>
> “The limitations should point out that a VPT like recipe would only work on RL domains where large-scale unlabelled data is accessible.”
>  - We’ve changed some language that hopefully makes this more clear. Thank you for the comment!

---

> > ### Comment · Reviewer_m9Gz · 2022-08-04
> > **Response**
> >
> > Thanks for the answers, there weren't major concerns anyways and I am happy that the authors have further elaborated on some of the design decisions. I believe this paper would be influential and should be highlighted at the conference.

---

### Official Review · Reviewer_RmjQ · 2022-07-11

**Rating:** 8
**Confidence:** 4
**Soundness:** 4 excellent
**Presentation:** 4 excellent
**Contribution:** 4 excellent

**Summary:**

This paper introduces Video Pre-Training (VPT), which is a new algorithm to learn foundation models for complex, long-horizon embodied agents in the popular Minecraft game. First, an inverse dynamics model is trained with human contractor data in Minecraft to predict keyboard and mouse actions from videos. Second, the inverse dynamics model is used to process large amounts of YouTube videos to obtain noisy action labels. Third, an agent imitates from the human players in YouTube by behavior cloning on the predicted actions. Finally, the policy can be further finetuned with RL on challenging tasks.

**Questions:**

The questions are listed in the "Weakness" section above to ask for more ablations.

**Limitations:**

The authors have addressed the limitations adequately.

**Strengths And Weaknesses:**

# Strengths

* Novelty: this paper proposes to use an inverse dynamics model learned from human contractor data to automatically label much larger amounts of YouTube data. The idea is simple, powerful, and novel.
* Performance: the VPT foundation models exhibit impressive zero-shot behaviors. The VPT model fine-tuned through RL is able to craft diamond tools, which is an extremely long-horizon task that takes up to 24,000 actions. Minecraft is a very challenging domain, so the performance is impressive and solid.
* It is surprising that keyboard and mouse action space can be learned effectively and works quite well. This is a promising direction for solving other challenging video games.
* This paper is well-written and easy to follow.

# Weaknesses

While the final performance of VPT is impressive, I hope to see more ablations studies:

1. On the effect of pretraining. From figure 9 we know that IDM can be quite data efficient. So what if the IDM is only trained on the house building contractor data, and then train BC on the labeled early game data as well as the house building data? This may tell us whether we need the "pretraining stage" in the paper.

2. On the effect of data filtering. The paper states that "With enough data, a large enough model, and enough training compute, a BC model trained on both unclean and clean videos would likely still perform well" - is there any concrete evidence on how much data filtering is worth in terms of model capacity, data size, and training budget?

3. On the sequence length of IDM. There is a 3D Conv to aggregate nearby frames (similar to frame-stacking). Why do we still need a long sequence (128) to train the IDM? Will the performance drop significantly with smaller sequence lengths?

4. On the model scalability of IDM. Figure 19 provide some scalability analysis of IDM on the "full" dataset. However, according to Figure 9, we do not need such large amount of data. So is such a large IDM still necessary? Some scalability analysis on smaller contractor dataset may be illustrative.

---

> ### Author Response · Authors · 2022-08-02
> **Response**
>
> Thank you for your thoughtful review. We’re glad you think that our method is novel, simple, and powerful, our results are impressive, and that the paper is well written! We will respond to each of your suggested ablations below, and we hope that after we include some of your suggested ablations along with additional clarification you will consider increasing your review score.
>
> “On the effect of pretraining. From figure 9 we know that IDM can be quite data efficient. So what if the IDM is only trained on the house building contractor data, and then train BC on the labeled early game data as well as the house building data? This may tell us whether we need the "pretraining stage" in the paper.”
>  - Figure 5 (right) shows an ablation where we show fine-tuning performance as we vary the amount of pretraining data used. The x-axis ranges from 0 epochs to 30 epochs of pretraining. The leftmost point (0-epochs) is the same as training from scratch on the target dataset (i.e. on the Early Game or Contractor House Building datasets without pretraining). Performance is lower across the board when doing no pretraining (0-epochs leftmost point) compared to full pretraining (30-epochs rightmost point), which is evidence that pretraining is helping.
>
> “On the effect of data filtering. The paper states that "With enough data, a large enough model, and enough training compute, a BC model trained on both unclean and clean videos would likely still perform well" - is there any concrete evidence on how much data filtering is worth in terms of model capacity, data size, and training budget?”
>  - Great question and we are interested as well. We’ve run an ablation comparing models trained on cleaned and uncleaned early game data (see Appendix section A.2.3 of the updated text). We ran this ablation in the regime with equivalent compute, so for the cleaned dataset which had about ~2000 hours of data we ran for 20 epochs, and for the uncleaned dataset which had about ~14000 hours of data we ran 2.8 epochs. In this regime we found it very beneficial to clean the data, which yielded a ~10x improvement in crafting of crafting tables and evidence of wooden item crafting versus none at all for the model trained on unclean data. We hope this provides some intuition as to how useful our data cleaning pipeline was.
>
> “On the sequence length of IDM. There is a 3D Conv to aggregate nearby frames (similar to frame-stacking). Why do we still need a long sequence (128) to train the IDM? Will the performance drop significantly with smaller sequence lengths?”
>  - Great question. At your suggestion we ran an experiment where we mask the attention matrix in the transformer layers such that only bands of varying numbers of neighboring frames (from 0 to 64) are included in the attention. So for instance, 0 neighbors would be the case where each transformer layer cannot attend to any neighboring frames and they are equivalent to MLPs with residual connections. We found that only the case of 0 neighbors was far worse than the architecture we use in the paper, and that each other configuration we tested (2-64) were comparable to the full 128 unmasked IDM used in the paper. We’ll add these results to the appendix.
>
> “On the model scalability of IDM. Figure 19 provide some scalability analysis of IDM on the "full" dataset. However, according to Figure 9, we do not need such large amount of data. So is such a large IDM still necessary? Some scalability analysis on smaller contractor dataset may be illustrative.”
>  - Figure 19 shows model scaling analysis of the VPT foundation model, not the IDM. We did not make this entirely clear from the caption and will update it to make it more clear.
>  - As to IDM model scalability, IDM training was a very small portion of the overall compute used in this project compared to foundation model pretraining. For this reason we did not investigate IDM model size thoroughly and only did minor tuning for accuracy and loss. On the other hand, IDM data scaling was much more important to us, which is why we provide ablations for this in Figure 9. We’ll add a comment saying as much in the paper to make all of this more clear.

---

> > ### Comment · Reviewer_RmjQ · 2022-08-09
> > **Good rebuttal! Consider adding discussion to a concurrent work**
> >
> > Thanks for the answers to my questions, they are all satisfactory!
> >
> > One minor suggestion: could you please discuss relations to a recent concurrent work, MineDojo: Building Open-Ended Embodied Agents with Internet-Scale Knowledge (https://arxiv.org/abs/2206.08853)? It seems very complementary, and readers from the community may benefit from the discussion in the final camera-ready draft of this paper.

---

### Meta-Review · Area_Chair_3MYT · 2022-08-25

**Recommendation:** Accept
**Confidence:** Certain

**Metareview:**

The authors have introduced Video Pre-Training (VPT), a semi-supervised learning approach that allows relatively small volumes of labeled data to train an inverse-dynamics model that is subsequently applied to predict the action labels associated with a far larger, unlabeled dataset. They then train an agent in a supervised regime with respect to these labels to achieve strong performance in Minecraft, which requires reasoning over very long time horizons.

Overall there is clear consensus among the reviewers that this paper is novel, technically sound and of broad interest to the NeurIPS community. The authors have also proactively engaged with reviewer feedback to improve manuscript clarity. I am confident in recommending this paper for acceptance.

**Award:**

No

---

### Decision · Program_Chairs · 2022-09-14

Accept